

# Glacier Image Velocimetry: an open-source toolbox for easy and rapid calculation of high-resolution glacier-velocity fields

Maximillian Van Wyk de Vries[1,2] and Andrew D. Wickert[1,2]

[1]Department of Earth & Environmental Sciences, University of Minnesota, Minneapolis, MN
[2]Saint Anthony Falls Laboratory, University of Minnesota, Minneapolis, MN

**Correspondence:** Maximillian Van Wyk de Vries (vanwy048@umn.edu)

**Abstract.** We present 'Glacier Image Velocimetry' (GIV), an open-source and easy-to-use tool for rapidly calculating high spatial and temporal resolution glacier-velocity fields. Glaciers' velocity fields reveal their flow dynamics, stability, and thickness. Obtaining widespread glacier-velocity measurements in the field is challenging and labour intensive. Recent increases in the availability of high-resolution, short-repeat-time optical imagery improve this, as persistent irregularities on the ice surface

allow us to use 'feature tracking' – an accidental form of 'particle image velocimetry' to obtain displacement fields, and hence, velocity over time. While these techniques have been used to calculate velocity fields for many glaciers, existing toolboxes can be expensive, complex or inflexible to use. GIV is fully parallelized, and automatically detects, filters, and extracts velocities from large datasets of images. Through this coupled toolchain and an easy-to-use GUI, GIV can rapidly analyse hundreds to thousands of image pairs on any modern laptop or desktop. We present four examples of how this model may be used: to

complement a glaciology field campaign (Glaciar Perito Moreno, Argentina), calculate the velocity fields of small (Glacier d'Argentière, France) and very large (Vavilov ice cap, Russia) glaciers, and determine the ice volume present within a tropical ice cap (Volcán Chimborazo, Ecuador). Fully commented code and a standalone app for GIV are available from GitHub and Zenodo.

## 1  Introduction

Satellite imagery revolutionized our ability to study changes in the surface of our planet. Satellite datasets now routinely support storm and drought evaluations (AghaKouchak et al., 2015; Klemas, 2009; Rhee et al., 2010), volcanic activity monitoring (Harris, 2013; Wright et al., 2002), and landslide-hazard analysis (Marc and Hovius, 2015; Metternicht et al., 2005; Tralli et al., 2005). In glaciology, remote sensing has enabled global glacier inventories (Earl and Gardner, 2016; Pfeffer et al., 2014)

as well as high-resolution elevation models and image mosaics of the Antarctic and Greenland ice sheets (Bindschadler et al., 2008; Howat et al., 2014; Hui et al., 2013; Porter et al., 2018). With temperatures consistently rising throughout much of the





globe, these images also provide an important temporal record of changes in ice extent and volume, as well as an effective tool for communicating these changes to the broader public(Stocker et al., 2013).

The use of imagery is not limited to mapping changes in glacial extent. The snowline on temperate glaciers, easily visible
from end-of-melt-season images, defines the equilibrium-line altitude, thereby delineating glacier accumulation and ablation areas (Bamber and Rivera, 2007; Rabatel et al., 2008; Yuwei et al., 2014). Identifying both seasonal and annual changes in snowline can provide important information about local winter precipitation, summer air temperatures and longer-term glacier mass changes (Bakke and Nesje, 2011).

Velocity measurements permit scientists to map glacier divides and drainage basins (Pfeffer et al., 2014; Davies and Glasser,
2012; Mouginot and Rignot, 2015) and track changes in surface melt production and accumulation (Mote, 2007; Sneed and Hamilton, 2007). Advancing techniques to remotely sense glaciers – and particularly their velocities – continues to provide new avenues to address key questions in ice dynamics and the future of glaciers under a changing climate (Rignot et al., 2011; Wal et al., 2008; Willis et al., 2018). Even the earliest glaciologists identified that glaciers may flow as viscous fluids (Bottomley, 1879; Forbes, 1840, 1846; Nye, 1952), and later that glacier surface motions are a complex interplay between
internal deformation, basal sliding, and deformation of subglacial sediments (Deeley and Parr, 1914; Weertman, 1957; Kamb and LaChapelle, 1964; Nye, 1970; Fowler, 2010). Sudden peaks in velocity may result from a sudden change in basal sliding, perhaps as the result of changing englacial hydrology. Long-term speedups or slowdowns may reflect climatic shifts or drainage reorganizations.

Deriving glacier velocities from satellite imagery is possible through an image-analysis technique known as 'feature track-
ing', 'image cross correlation', or 'particle image velocimetry'. The latter term, 'particle image velocimetry', describes a well-established technique in fluid dynamics, typically involving the use of a high-speed digital camera to track the motion of tracers within a fluid in a laboratory setting (Buchhave, 1992; Grant, 1997; Raffel et al., 2018) . These ideas were first carried over to the field of glaciology by Scambos et al. (1992) to evaluate the flow velocity of a portion of an Antarctic ice stream. Since that time, the increasing abundance and availability of imagery has facilitated expanded use of feature tracking-based
velocimetry techniques. With the release of the full Landsat data archive and launch of Sentinel-2, 10-30 m pixel resolution imagery of any given location is now available at sub-weekly repeat coverage intervals. A number of studies use this exceptional potential to map the velocity of the major ice sheets as well as many glaciers around the world (Gardner et al., 2018; Millan, 2019).

Prior to the advent of remote sensing, spatially distributed measurements of glacier flow velocities required lengthy field
campaigns (Chadwell, 1999; Hooke et al., 1989; Mair et al., 2003; Meier and Tangborn, 1965). Nowadays full 2D flow-velocity maps may be readily calculated from a variety of optical and radar-based satellite imagery (Heid and Kääb, 2012b). For this toolbox we focus on optical imagery products due to their ease of access, limited need for pre-processing and high spatial and temporal resolution (Darji et al., 2018; Drusch et al., 2012; Heid and Kääb, 2012b, a; Kääb et al., 2016).

A number of tools exist to derive displacements from imagery, as partially reviewed by Heid and Kääb (2012a); Jawak et al.
(2018) and Darji et al. (2018). Table 1 presents a non-exhaustive list of software packages available online. Our objective with the 'Glacier Image Velocimetry' (GIV) toolbox presented here is to provide an easy to use, flexible, and efficient tool that can



be used to derive high spatial resolution and monthly temporal resolution surface-velocity maps of any glacier. The following section will run through the basics of image feature tracking techniques and advances built into GIV.

**Table 1.** List of existing codes and toolboxes that may be used for feature tracking, and associated references.

| Toolbox | References |
|---|---|
| ImCorr (Image Correlation) / ampcor | Scambos et al. (1992); Millan (2019); Nagy and Andreassen (2019) |
| CIAS (Correlation Imaging Analysis Software) | Kääb and Vollmer (2000); Heid and Kääb (2012a) |
| ImGRAFT (Imagining Georectification and Feature Tracking) | Messerli and Grinsted (2015) |
| Cosi-Corr / ENVI | Leprince et al. (2007a, 2008); Scherler et al. (2008) |
| fourDvel | Minchew et al. (2017) |
| vmap | Shean (2019) |
| Auto-RIFT | Gardner et al. (2018) |
| SendIT | Nagy and Andreassen (2019); Nagy et al. (2019) |
| Cryosphere And Remote Sensing Toolkit (CARST) | Willis et al. (2018); Zheng et al. (2019a, b) |
| PIVlab | Thielicke and Stamhuis (2014) |
| matpiv | Sveen (2004) |

## 2   Methods and model setup

The fundamental idea of feature tracking is based on techniques used to co-register images: the properties of two images are compared in order to identify the best-fit location of one image within the other (Messerli and Grinsted, 2015; Scambos et al., 1992; Thielicke and Stamhuis, 2014) . In feature tracking, including in GIV, individual images are broken down into a grid of smaller images (referred to as 'chips'). We compare each individual 'chip' from the first image (I1) to the corresponding portion within a second image (I2), and find the best matching portion of I2. If no displacement has occurred between the two 65   images, the best-fitting portion of I2 will have the same location as the original 'chip' on I1 (excluding any georeferencing or distortion-related errors). However, if any motion has occurred between the two images, the corresponding best matching 'chip' within I2 will be displaced relative the original location within I1. We may then determine the bulk displacement in pixels between the original I1 'chip' and best match I2 'chip'. The correlation coefficients between the original chip and surrounding area within I2 are also calculated. This allows a Gaussian curve to be fit to this grid in order to determine the peak location 70   at sub-pixel accuracy. Repeating this routine for every chip within the original image allows a fully distributed 2D surface velocity field to be derived.

When initially developed for use in laboratory-based fluid dynamics, the camera, lighting, and tracer-particle conditions were all closely constrained (Grant, 1997; Raffel et al., 2018). On glaciers, features change over time as crevasses open and close, snow drifts, and ablation exposes new surfaces. In addition, the satellite may acquire imagery from slightly different locations



and angles with each pass, and lighting conditions depend strongly on the time of day and year, as well as local weather
conditions (Berthier et al., 2005; Kääb et al., 2016). This complexity raises additional problems in the use of this technique
for deriving glacier velocities, and makes it entirely unusable in some cases (e.g. images too far spaced in time or flow too
rapid for glacier surface to retain any coherence). These problems, however, are not insurmountable, and can be mitigated
though a combination of image pre-filtering, comparison between adjacent velocity maps, and outlier filtering. The Glacier

Image Velocimetry toolbox makes full use of all of these approaches, with a particular emphasis on reducing the uncertainty of
individual velocity maps through the use of large datasets. Figure 1 presents the overall model setup and order of operations.

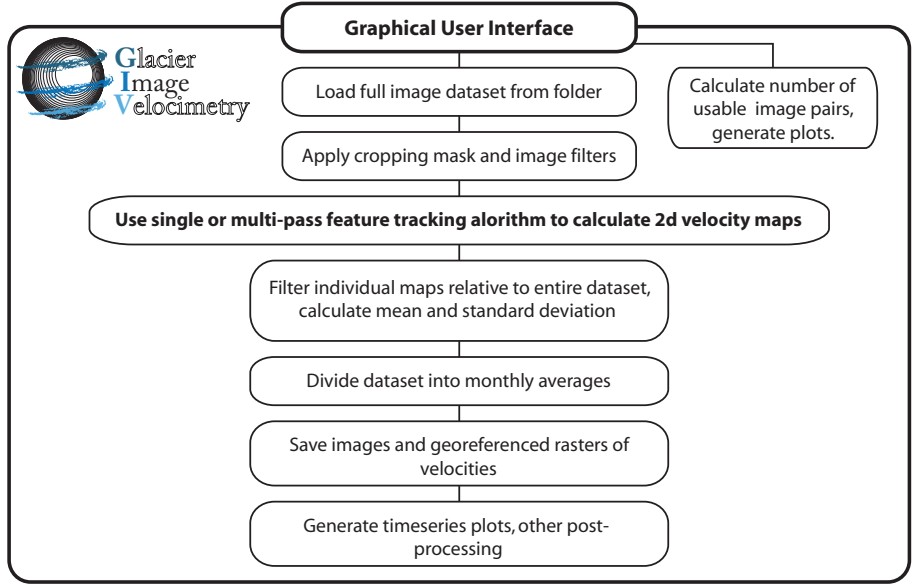

**Figure 1.** Flowchart describing the Glacier Image Velocimetry toolbox design. Users enter parameters into fields in the user interface. All
subsequent steps are automatically performed by the toolbox.

## 2.1   Model pre-processing

Prior to running the feature-tracking algorithms, the images are first loaded into the workspace and filtered. The user interface
will prompt the user to enter the coordinates of the images (minimum and maximum latitude and longitude), and to select a

given folder in which the images are stored. These images must be .png or .jpg files of the area, with each file name being the
date of image acquisition (in yyyymmdd format). GIV will then extract the dates from the file names, calculate time between
images, and load the raw image data into an array for further processing. The user also inputs a modified image with glaciers
of interest converted to pure white (RGB 255,255,255). This image is loaded by GIV and converted into a binary mask with
areas within (1) and outside (0) the computational region. The size and resolution of images are also automatically calculated

and resampled to a common resolution, such that images from different satellites may be combined into the same dataset.



Following this, GIV filters the images following user-defined settings. GIV includes a range of filters in order to reduce the effect of unwanted noise (e.g. clouds and shadows) and emphasize trackable features (e.g. crevasses, snowdrifts, supraglacial debris). In particular we include high-pass, Sobel, and Laplacian filter options to emphasize short-wavelength features and edges, as well as intensity-capping and contrast-limited histogram-equalization filters to improve image contrast (Gardner et al., 2018; Sveen, 2004; Thielicke and Stamhuis, 2014). We also developed a 'near anisotropic orientation filter' (NAOF), which in most cases produces the highest number of correctly tracked velocity 'chips'. We define this filter as:

$$I_f = \sum_\alpha \mathrm{Re}\left[\exp\left(i \times \arctan 2(I_o * \alpha, I_o * R[\alpha]]\right)\right] \tag{1}$$

With $I_f$ the filtered image, $I_o$ the original image, $\alpha$ representing four different convolution matrices oriented at 45 degrees from each other using the 8 adjacent pixels, $\mathrm{Re}[x]$ representing the real portion of complex number x, $\arctan 2(x,y)$ representing the four-quadrant arctangent (also called the two-argument arctangent), $x * y$ representing a two dimensional matrix convolution, and $R[x]$ representing a 90 degree matrix rotation. This filter works by summing differently angled orientation filters together in order to recover a 'pseudo-feature' with the same location as the original feature, but with an increased contrast between the feature and the background, and homogenized magnitude (Fitch et al., 2002; Kobayashi and Otsu, 2008) . Information on absolute pixel color magnitude is discarded, with only information on color gradients preserved. A similar result may be obtained by convolving the original image with a single symmetrical convolution matrix, but this also normalizes all features to a single magnitude and results in a larger number of false matches. The NAOF filter has the advantages of (a) strongly increasing the contrast between features and background; (b) removing contrast differences between clouded, shadowed, and clear areas; and (c) preserving feature uniqueness. Figure 2 shows examples of how this filter is able to recover features from otherwise unusable images. Many glaciated areas remain partially cloud covered and shadowed for much of the year, so being able to recover partial velocity fields from these images can greatly increase the size of potential datasets. Note that no amount of filtering can improve certain images, such as those in which cloud cover is too thick for the surface to be visible.

## 2.2 Velocity calculations

Two main methods exist to derive displacements from an image pair. The first involves only a single pass across the images, and the second involves multiple passes with gradually reducing window sizes (Raffel et al., 2018; Thielicke and Stamhuis, 2014). Single-pass methods have the advantage of generally being faster at coarse resolutions and are less at risk of smearing one erroneous value over a larger area. Multi-pass methods are generally more accurate and make use of the reduction in window size to increase signal-to-noise ratio. Both methods are integrated into GIV. The single-pass method is based on a function from ImGRAFT (Messerli and Grinsted, 2015) and the multipass method was edited based on the matpiv toolbox (Sveen, 2004). Both functions have been tested in a number of previous studies, with matpiv being used extensively in fluid-dynamics research (e.g., Lee et al., 2017; Oertel and Süfke, 2020; Sveen and Cowen, 2004; Sveen, 2004).





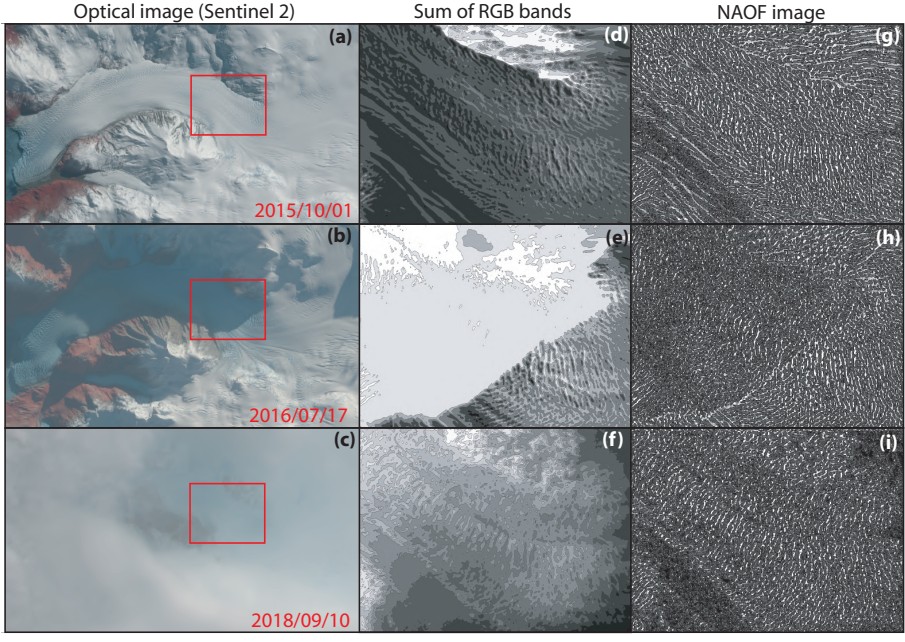

**Figure 2.** Comparison between raw optical images, band summed images and NAOF filtered images for a clear image (a, d, g), a heavily shadowed image (b, e, h) and a cloudy image (c, f, i). Note how despite the complexity, the NAOF images recover a clear and easily traceable feature pattern on the surface of the glacier that is suitable for obtaining velocities. The shadow line leaves an artefact in h, but is a marked improvement on the lack of features in the shaded area in e. Images from Sentinel-2.

The core of both the single and multipass methods involves converting each image chip to the frequency domain using a fast Fourier transform (FFT) algorithm, calculating the correlation coefficient with surrounding areas within a given search window, and converting the resulting similarity matrix back to the spatial domain with an inverse FFT (IFFT). This step is repeated on each chip within the original image, and is the most computationally expensive of the entire process.

GIV is written in MATLAB. Despite being a high-level interpreted programming language, MATLAB performs FFT calculations using precompiled C and Fortran bindings for the FFTW library (Frigo and Johnson, 2005, 1998). Due to this being the rate-limiting step in feature tracking calculations (>90% of computation time in most cases), such code may be written in MATLAB with few performance issues relative to other programming languages.

As the feature-tracking correlation between two images inherently requires a large number of FFT and IFFT operations, this step has limited potential for further optimization. Computation time may instead be decreased by deriving displacement fields from different image pairs in parallel rather than in series. This requires a slightly different code design: First, GIV detects the number of physical cores on the user's computer and starts a parallel pool. It then decomposes the full sequence of image pairs into sub-sequences the size of the number of cores. Finally, it distributes each sub-sequence across the cores in the computer to be computed in parallel. Figure 3 shows the increase in computation speed with number of cores used in different scenarios. This enables large datasets to be processed more rapidly, even on standard laptop and desktop computers.


GIV may also calculate velocity maps for pairs of non-consecutive images, which we refer to as "temporal oversampling", resulting in much larger final datasets. The user inputs maximum and minimum temporal separations for image pairs, and GIV extracts all suitable pairs, including those that are not consecutive. For a dataset of n images, this theoretically enables a total

of (n2-n)/2 image pairs (or 19,900 image pairs for a 200 satellite image time-series). For heavily clouded datasets this also has the advantage of increasing the likelihood of forming cloud-free image pairs.

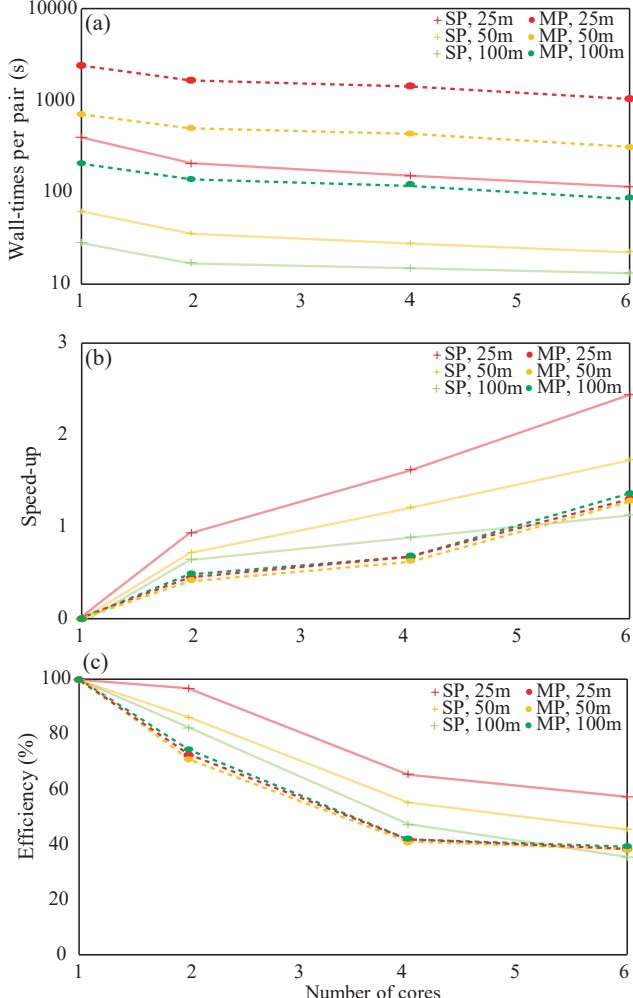

**Figure 3.** MP = Multi-pass; SP = Single-pass. Test conducted on a 12-image dataset of 10-m resolution, 1.7 million pixel images of Amalia Glacier, Chile using a Dell XPS 15 laptop (2×16GB DDR4 2666 MHz memory, 6-core Intel i7-8750H 2.20 GHz processor). In all cases, parallelization decreases runtime, and going from one to two cores improves runtime by 1.4–1.9×. Fine-resolution multi-pass runs usually yield the best velocity fields, and (b) shows that these benefit from the largest speed-up when parallelised.

Apart from some scenarios and locations such as surges, spring speedups, and the margins of ice streams, glacier velocity gradients vary gradually in both space (low lateral velocity gradients) and time (low acceleration). Therefore, the accuracy of





individual velocity measurements can be evaluated by comparing them to their immediate neighbours in both space and time.
Sudden jumps in either most likely represent erroneous velocities due to mismatches within the feature-tracking algorithm.
This property is used in the GIV toolbox to improve the final velocity maps through the following workflow:

Firstly, GIV filters each individual velocity map through user-prescribed limits on velocity and flow direction, as well as
outlier detection functions. This finds values that differ by more 50% from their immediate neighbours (4 surrounding cells)
and 200% from the mean of their larger local area (25 surrounding cells), removes these outlier values, and interpolates across
these now-empty pixels using the remaining values. Secondly, GIV calculates the mean, standard deviation, median, minimum,
and maximum velocities across the full dataset. It then compares each individual value to the mean value at that location for
the entire dataset. Any values more than 1.5 standard deviations away from this mean are considered outliers. This process
is carried out both for the velocity and flow-direction grids, and only values within the threshold for both velocity and flow
direction are conserved. This provides an additional check, as erroneous values are unlikely to coincidentally match both
the velocity and flow direction. Finally, the entire dataset may be smoothed and interpolated in space and/or time and space
according to the user's choices. This allows missing values at one timestep to be infilled from neighbouring times if the dataset
is smooth enough to allow it. In addition, the displacement of each image pair may be normalized to the displacement of user
defined stable ground to correct for systematic georeferencing errors.

Variable satellite repeat intervals and the exclusion of entirely clouded or otherwise unusable images lead to unevenly
spaced velocity timeseries that are more difficult to interpret. In order to reduce this challenge, GIV includes a function that
automatically averages the data and resamples it to monthly intervals. This is easy when all individual velocity maps cover
periods of less than one month and do not overlap between months, but becomes more complex when they do. In many cases,
image pairs with the shortest lag times (<7–10 days) are excluded because displacement over such a short time may be much
smaller than offsets due to distortion and/or georeferencing errors. For the slowest-moving glaciers, this lower bound may be
extended to several weeks or months. Lag times as long as the available imagery time series may be used so long as the surface
of the glacier retains coherence in the image pairs.

GIV can determine monthly values by averaging across all image pairs that overlap with a given month. However, this will
likely produce an artificially smoothed dataset due to the influence of velocities measured across the boundaries of months.
In order to make use of longer lag-time pairs, we develop an iterative strategy for calculating monthly values. In the first
place, GIV takes a weighted mean of all velocities covering that month to make an initial guess at monthly velocities. The
weighting parameter is determined by the proportion of the individual map contained within the given month, so for instance
a velocity entirely within one month will be weighted 1 and one spread evenly over four months will be weighted 0.25. This
initial estimate is then used to iterate between monthly averages and raw data values, with raw values covering more than one
month split into monthly values by subtracting the previous iteration's estimate of monthly averages from them (Figure 4). This
procedure may extract average monthly velocities even for months lacking any data. Outlier detection and maximum velocity
filters are implemented to prevent small errors in the raw data from being accentuated by the iterations, but this may also lead to
loss of data if too large a proportion of the initial dataset is inaccurate. Due to this limitation, the iterative calculations may not
be adapted to some noisy datasets, for which the loss of temporal resolution by simple averaging will be preferable. Monthly

averaging is performed as a post-processing step, and so may be repeated without the need to recalculate any raw velocity
maps. Time series may also be generated from the raw data if monthly averaging is not desirable.

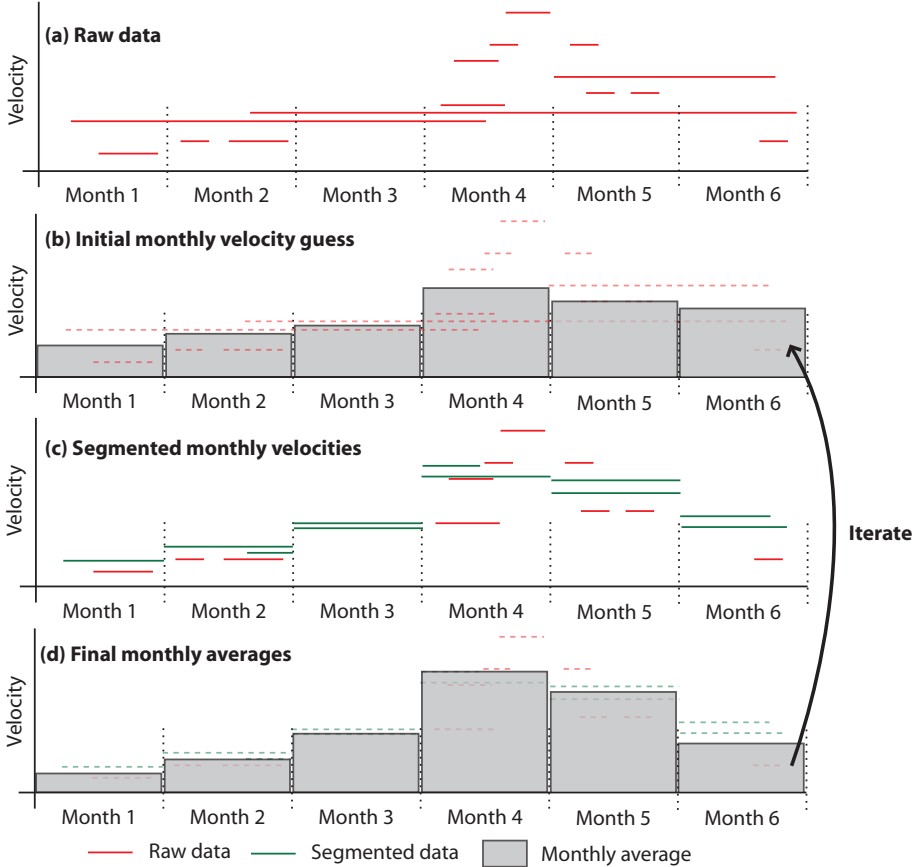

**Figure 4.** Schematic description of the techniques used to derive monthly velocities. The raw data (a) are combined in a weighted average
to make an initial guess of monthly averages (b). The monthly averages are then used to segment longer time period velocity maps into their
different monthly contributions (c). These are used to recalculate the monthly averages (d). Finally, GIV iterates over steps b–d for a number
of times (e.g. 10) provided by user inputs. Note that an estimate may be made for the average velocity in 'Month 3', despite this month
having no imagery available.

As a final step, GIV will automatically georeference the velocity grids and save .tif files to the user's computer. The toolbox
also contains mapping tools that allow automatic generation of publication-quality images of the velocity and flow-direction
maps (figure 5). In the following section we will examine some case studies of real glaciers and scenarios for which this model
may be useful.





## 3  Results and Examples

Ice-velocity measurements supply essential information for studies of glacier dynamics, thickness, subglacial hydrology, and mass balance. With its GUI-based inputs and potential for parallelization, GIV can calculate a monthly velocity field for any glacier around the world with only a few hours of work. As such, it may also be run alongside field-based expeditions in order to understand the current conditions of the glacier and aid in instrumentation positioning.

We present four case studies. The first is of Glaciar Perito Moreno, where we use GIV to determine the displacement of automated ablation stakes in conjunction with fieldwork in Spring 2020. The second is Glacier d'Argentiere, a small and well-studied valley glacier located in the French Alps. The third is the Vavilov ice cap, located on October Revolution Island, in the Arctic Ocean off the mainland Russian coast, whose western outlet glacier is now surging into the ocean. We validate PIV against published results (Zheng et al., 2019b) using another image-based ice-velocity tool, CARST (Zheng et al., 2019a). Finally, we compute ice-flow velocities across Chimborazo ice cap in Ecuador, and use these to invert for ice thickness.

### 3.1  Field-campaign support: Glaciar Perito Moreno and the Southern Patagonian Icefield

A team from the University of Minnesota installed 3 automated weather stations and 3 automated ablation stakes near the southern flank of Glaciar Perito Moreno in order to better understand the local conditions of this glacier and construct temperature-index and energy-balance models for glacier ablation. We installed the automated ablation stakes, based off of designs by Wickert (2014) and Wickert et al. (2019), and tested by Saberi et al. (2019) and Armstrong and Anderson (2020), for 20 days between the 23rd of February and 14th of March, 2020. In slowly flowing glaciers, ice flow may be largely neglected when considering equipment recovery. In rapidly flowing glaciers such as Perito Moreno, however it may be relevant to consider the movement of the glacier when planning equipment recovery. This is particularly relevant where intense crevassing makes both access and visibility difficult. Figure 6 shows how different positioning decisions may influence ease of recovery: ablation stakes installed in position PM1 will move tens of metres towards the centre of the glacier in less than a month, whereas stakes in position PM3 will move less than 5 m towards the glacier flank. In our survey, stakes were installed around position PM3 for ease of access.

Figure 6 (b) also presents the case of Glaciar Europa, which drains the adjacent portion of the Southern Patagonian Icefield in Chile. We also derived the mean velocity field of this glacier over the past 3 years using Sentinel-2 imagery (195 image pairs). GIV velocity measurements reveal that the central portion of Glaciar Europa at its outlet flows nearly 10,000 m/yr. If an ablation stake were installed in this area (point EU1), it would be displaced almost half a kilometre over the course of a 20-day survey. If it were instead placed at an alternative location 1 km to the West (EU2) it will be displaced only 20 metres in the same time period. This is an extreme case, and the flow speeds of most glaciers are orders of magnitude slower, but nonetheless reflects a situation in which deriving velocity fields would aid the success of a glaciological field campaign.



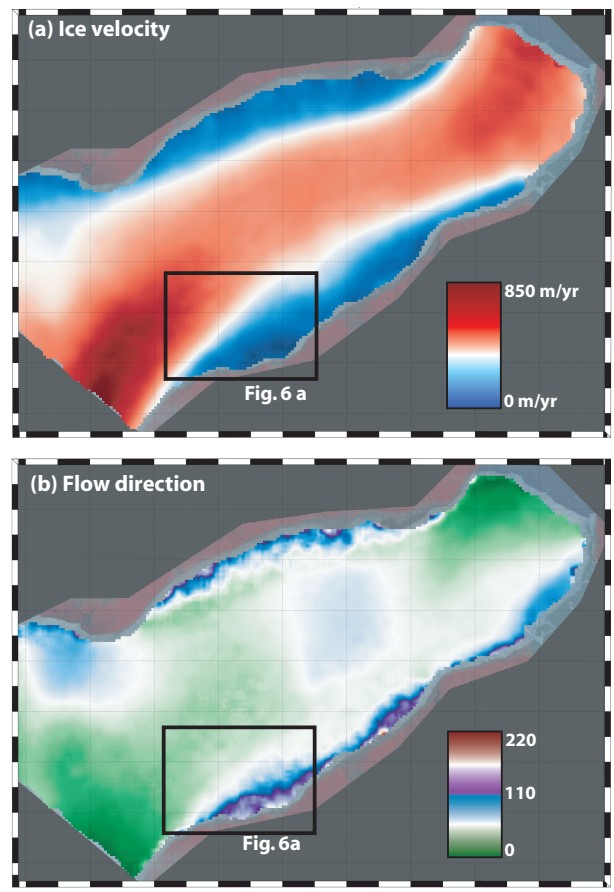

**Figure 5.** Mean flow velocity (m per year) (a) and direction (degrees) (b) of Glaciar Perito Moreno, Argentina for the first three months of 2020. Figure panels automatically generated from GIV, labels have been added and the color bars moved.

## 3.2 Valley-glacier velocity distribution: Glacier d'Argentière

In order to evaluate the effectiveness of GIV on smaller glaciers, we calculate a velocity field for a well-studied valley glacier in the Mont Blanc massif, Glacier d'Argentière (Benoit et al., 2015). We download one year worth of Sentinel-2 data (March 2019 – March 2020), containing over 1000 image pairs. These images are then used to derive a 25-m resolution mean-ice-velocity map, shown in Figure 7. The sparsity of features transverse to flow direction on Glacier d'Argentière make it difficult for feature-tracking methods to calculate velocities. Nevertheless, the resulting flow-velocity map is comparable to those derived using a SPOT satellite image pair from 2003 (Berthier et al., 2005; Rabatel et al., 2018), SAR and ground based photogramme- try (Benoit et al., 2015), and a different feature-tracking routine based on a modified version of ampcor (Millan et al., 2019). The velocity map highlights accelerated ice flow at the terminus icefall and on the steep tributary glacier to the SW of the main trunk (Figure 7). Main-trunk velocities are on the order of 45–70 m/yr, slightly slower than Berthier et al. (2005)'s SPOT



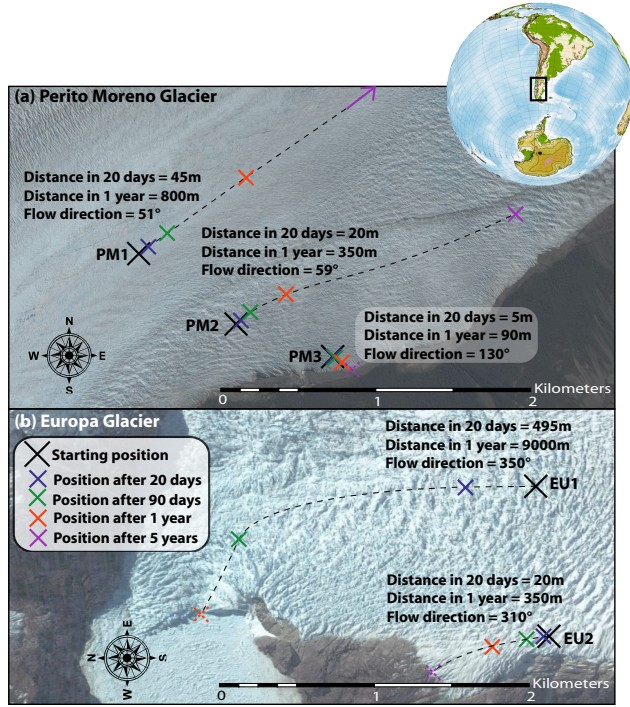

**Figure 6.** (a) Position of a point on Glaciar Perito Moreno with time for three different starting locations within 2 km of the glacier's southern margin. At PM1, ice speeds reach 800 m/yr and any equipment will be rapidly displaced. At PM3 ice-flow speeds are $< 100$ m per year and oriented towards the valley edge. (b) Identical plot for two points on Glaciar Europa. Any equipment installed at EU1 will be displaced several kilometres and lost to calving in less than 6 months. Imagery © Google Earth

values but in line with Benoit et al. (2015)'s values. Our values represent the mean over an entire year, including the slower winter velocities captured by Berthier et al. (2005). It is also possible that glacier thinning has reduced its flow velocity, but sufficient data to evaluate this do not exist.

### 3.3    Validating GIV by observing Vavilov ice cap dynamics

#### 3.3.1    Mapping ice surge

Arctic land-ice melt has contributed more than 20 mm to global sea level rise since the 1970s (Box et al., 2018). Most of these large glaciers and ice caps remain remote and difficult to access, and high spatial and temporal resolution surface velocity maps provide one important tool to assess their response to changing environmental conditions.

    The Vavilov ice cap is a 1700 km$^3$ ice cap located on October Revolution Island in the Severnaya Zemlya archipelago, located in the Russian high arctic (Bassford et al., 2006). Until the 2010s, the Vavilov ice cap exhibited surface velocities of 235    only a few tens of metres per year, typical of many cold-based high-arctic ice masses. In 2013, a large portion of the marine-terminating western flank surged, with the ice front reaching more than 10 km beyond its prior grounding line by 2016 (Willis





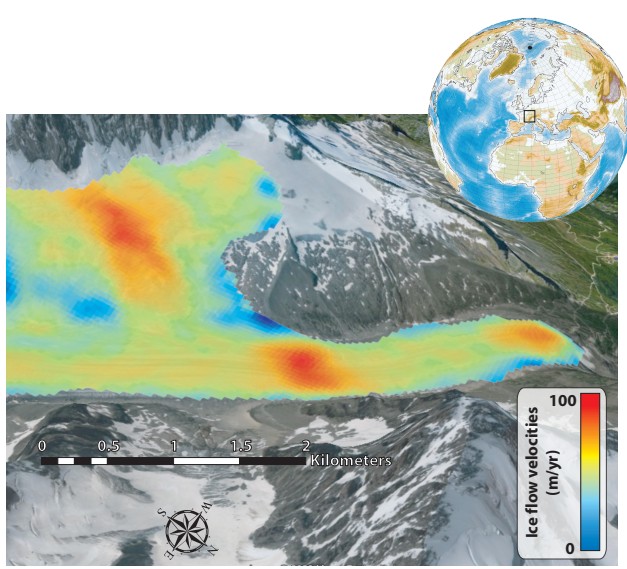

**Figure 7.** Perspective view of mean flow velocities of Glacier d'Argentière, France over the period 03/2019–03/2020. Imagery © Google Earth, scale for near margin of glacier.

et al., 2018; Zheng et al., 2019b). This sudden shift in ice behaviour was not accompanied by any dramatic climatic shift, and the exact triggers are a matter of active debate (Willis et al., 2018, and references therein). Willis et al. (2018) proposed that the dramatic acceleration is related to the ice cap overriding weak marine sediments in the Kara Sea, which can deform easily

and substantially increase ice velocity. The ice cap margin is also no longer frozen to bedrock, leading to associated removal of resistive stresses at the ice front (Willis et al., 2018). Rapid ice flow initiates a set of internal feedbacks to further increase ice velocity, including strain softening of thie ice itself; shear heating that produces meltwater, capable of reducing the effective normal stress of the ice and hence its friction against the bed; and potential infiltration of this water into the bed material, increasing its deformability (Willis et al., 2018, and references therein). With no direct data on subglacial conditions prior to

or during the surge, the exact processes involved remain difficult to reveal. We may, however, monitor surface ice velocities to examine the ongoing changes in ice dynamics.

Visible-band imagery from the Vavilov ice cap is available only for summer months (March to September) due to darkness during the high-latitude boreal winter. We use GIV to derive a 100-m resolution ice-velocity map of a 365-km$^2$ area of the Western flank of the ice cap using the entire Sentinel-2 archive (beginning in 2016). Figure 8 (a) and (b) present two average

yearly velocity maps for the apex of the surges in 2016 and 2019. Panels (d) and (e) present timeseries of monthly velocities over the period from March 2016 to March 2020 at the locations shown in figure 8(c).

Velocities of the centreline points converge over the time period considered: Although the velocities near the ice front decrease from the 2016 peak (red, orange and green circles), velocities of regions most distant from the coast show a steady increase (purple points). The central portion of this newly formed outlet glacier shows distinct late-summer accelerations in

both 2018 and 2019, reaching around double the spring and early summer rates and rapidly decaying (figure 7(d)). Within the


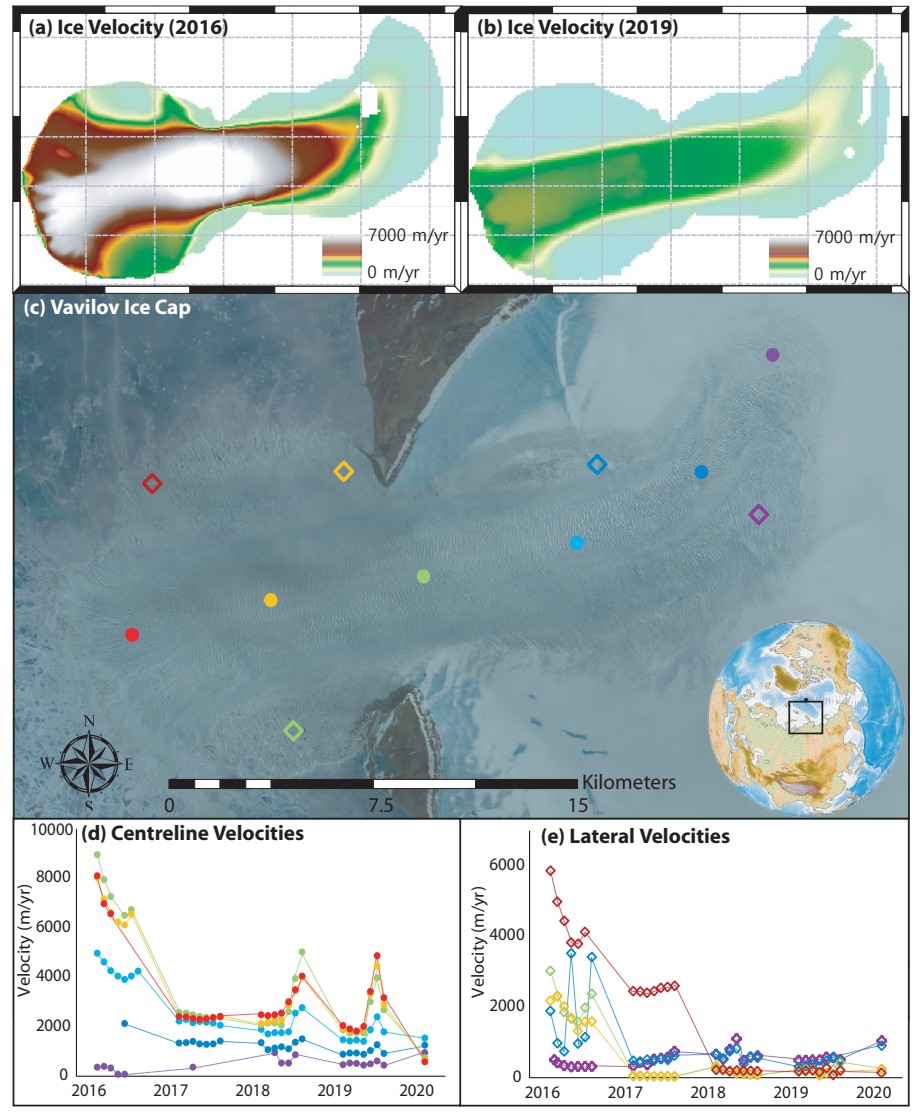

**Figure 8.** (a) and (b) present 100-m-resolution annual mean velocities for the western outlet glacier of the Vavilov ice cap. (c) displays a 2019 Sentinel-2 image showing the main features of this outlet and the locations used to derive monthly timeseries. (d) and (e) present monthly resolution velocity timeseries along the glacier centreline and flanks using Sentinel-2 imagery.

newly formed western frontal lobe extruded beyond the prior grounding line, flow has concentrated into a single branch with well-developed shear margins separating a central region with rapid ice flow from slow-moving lateral portions (Zheng et al., 2019b).

Extraction of high-resolution ice velocities in this region using GIV confirms Willis et al. (2018) and Zheng et al. (2019b)'s findings that the western portion of Vavilov has entered into a new fast-flow regime. The late summer velocity peaks in both





2018 and 2019 may shed some light on the driving forces behind this acceleration if associated changed in climatic, ice surface or ice basal conditions are detected. Ongoing monitoring will help to determine whether a similar peak occurs in 2020 or any following years, and can be performed in near real time using GIV.

### 3.3.2 Method validation

We compare our GIV-derived results against a velocity map of the front of the western outlet glacier generated by Zheng et al. (2019b) using CARST (Zheng et al., 2019a). Zheng et al. (2019b)'s velocities were generated based on a single Landsat 8 pair dated 2017/05/06 and 2017/05/22. We compared the ice-surface velocity magnitude calculated from this pair to the May 2017 average velocity map generated from Sentinel-2 imagery using GIV through the approach described above. We georeferenced the two velocity maps using the glacier margins and other notable features. The difference map (Fig. 9(a)) displays the highest

amplitude anomalies along the margins of the central high-velocity band. Differences between the GIV- and CARST-derived velocity maps are normally distributed, with a mean difference of $-16$ m per year (Fig 9(c)). This mean difference is $\leq 1\%$ the total velocity across the majority of the glacier surface (Fig 9(b)). In this region of the glacier surface, the annual variability in ice-surface velocities is on the order of several hundred metres per year (Fig 8(d) and (e)), and this difference between our results using GIV and those of Zheng et al. (2019b) could plausibly result from the slightly different dates covered or differing

image resolutions (10 m for Sentinel-2 compared to 15 or 30 m for Landsat). The high-magnitude difference bands on either side of the fast-moving central region may also result in whole or part from georeferencing errors in GIV, in CARST, or in our work to georeference these two velocity maps to one another.

### 3.4 Ice-thickness inversions and tropical glaciers: Chimborazo ice cap

There are many glaciers for which ice thickness measurements would be useful, but traditional radar or borehole techniques

are too challenging or expensive to apply. In these cases, we may combine our remotely derived ice-surface velocities with knowledge about ice-flow mechanics to estimate ice thickness and volume (Gantayat et al., 2014; Farinotti et al., 2019). Where no data are available, these approaches can provide a physics-based first estimate. Where even one or a few data points are available, these can help to calibrate ice-flow parameters and perform a physics-based extrapolation of local field measurements, resulting in a spatially distributed measure of ice thickness – and hence, volume.

Many tropical glaciers and ice caps have limited or no ice-thickness information (Thompson et al., 2011). These are important water sources to millions of people (Bury et al., 2011; La Frenierre and Mark, 2017; Chevallier et al., 2011). Vergara et al. (2007) estimated the economic cost of glacier retreat on water use to be in the hundreds of millions of U.S. dollars, and the impact on Peru's electrical utility to be $\sim 1.5$ billion. Accurate and spatially-distributed estimates of ice thickness can therefore support practical decision making in the tropical Andes.

Chimborazo is a 6268 m high stratovolcano in Ecuador capped with an ice cap and 17 outlet glaciers. On Chimborazo's north-eastern flank, glacier meltwater drives nearly all of the discharge variability, and the disappearance of the prominent Reschreiter Glacier could decrease the discharge of the watershed's outlet stream by up to 50% (Saberi et al., 2019). Due to its high elevation and steep slopes that are unstable in regions of recent ice retreat, the glaciers on Chimborazo are difficult to



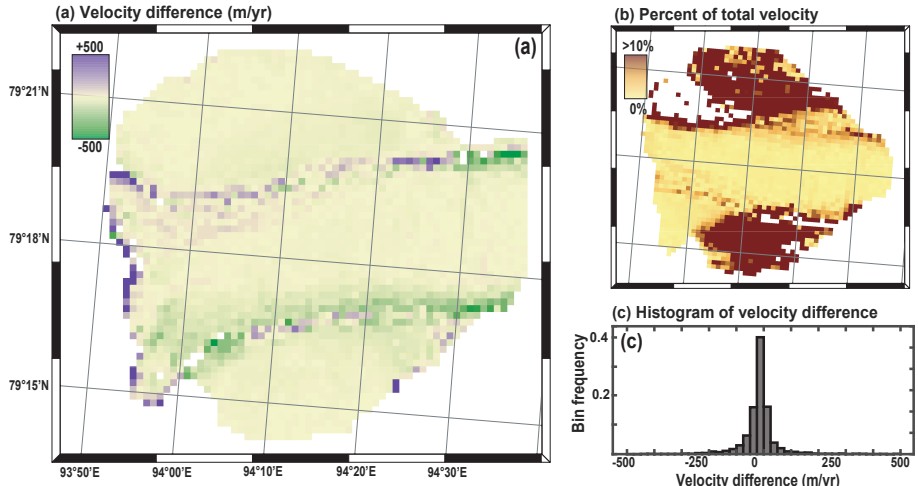

**Figure 9.** Comparison between Zheng et al. (2019b)'s velocity maps for Vavilov ice cap (Pair037_20170506_20170522) with results from GIV (May 2017 average). a) shows a difference map between the two results (Zheng et al. velocity minus GIV velocity), b) shows what percentage of the total velocity this difference represents (absolute value of the difference shown in a) divided by GIV velocity), and c) is a histogram of the difference values. The mean difference between the two velocity maps is less than 20 m per year, or less than 1% of the total velocity for much of the area.

survey (Saberi et al., 2019). Ice cores have been collected at the summit plateau in 1999–2000, the longest of which extends

54.4 m to the glacier bed (Schotterer et al., 2003; Ginot et al., 2010).The general data sparsity but existence of a single point for benchmarking, combined with its water-supply importance for the surrounding communities (La Frenierre and Mark, 2017), makes it an appropriate target for an initial ice-thickness inversion using GIV.

### 3.4.1    Inverting surface velocity for ice thickness

Glacier motion occurs though a combination of internal deformation, basal sliding and subglacial sediment deformation. Ice

flows under its own weight, and the rate of internal deformation is a function of the thickness of the ice. Ice-surface velocities $u(H)$ may be written as:

$$u(H) = u_{d,H} + u_s + u_t. \tag{2}$$

Here, the ice thickness $H$ denotes that the velocity is evaluated at the ice surface, $u_{d,H}$ is the surface velocity produced by internal deformation alone, $u_s$ is the rate of basal sliding, and $u_t$ is the component of glacier velocity produced through till

deformation.

We first work to remove terms from this expression. The glaciers on Chimborazo flow over bedrock, thus $u_t$ should be at or near zero. Glacial sliding requires warm-based ice and can be enhanced by water pressure (e.g., MacGregor et al., 2000). Meltwater has been identified near the summit of Chimborazo, where coring attempts in December 2000 were disrupted by a layer of waterlogged ice at a depth of 28 m (Schotterer et al., 2003). However, surface melt was exceptional in the

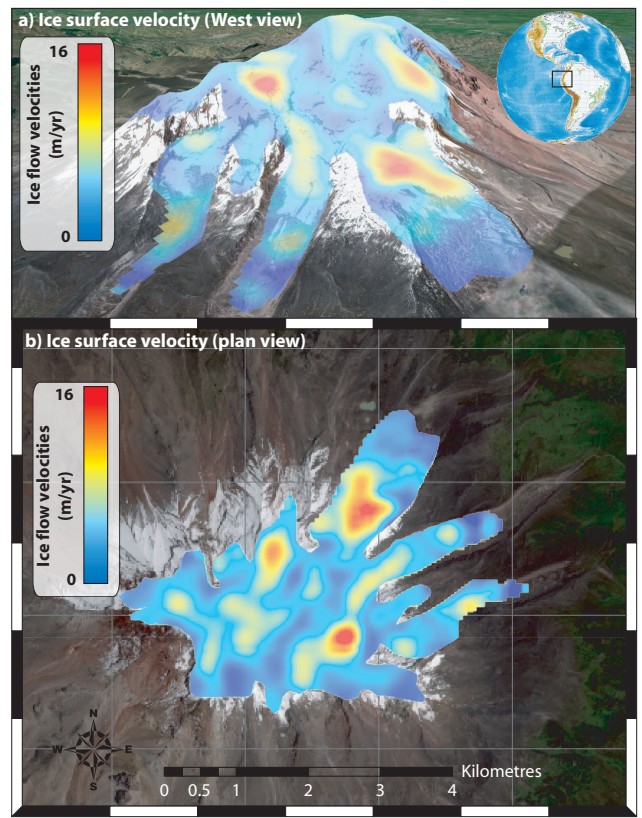

**Figure 10.** Ice surface velocity map for the Chimborazo ice cap calculated with GIV. Imagery © Google Earth and Sentinel-2.

years 1999–2000 due to tephra fallout from the nearby eruption of Tungurahua (Schotterer et al., 2003; Ginot et al., 2010). The $0°C$ isotherm was estimated at 5050 m using multiple field based temperature sensors (La Frenierre and Mark, 2017). This is consistent with a $0°C$ isotherm of 4922 m calculated from from two weather stations in the vicinity of Chimborazo (La Frenierre and Mark, 2017; Saberi et al., 2019): Boca Toma (W078.7508°, S01.4482°, elevation 3899 m) and Reschreiter camp (W078.7741°, S01.4459°, elevation 4355 m). This suggests that most or all of Chimborazo's ice cap is geographically

above and thermally below the 0 degree isotherm, and therefore is composed of cold-based ice that cannot slide. As a result of the hard bed and mostly cold-based ice, we approximate that internal deformation alone sets glacier-surface velocity:

As a result of the hard bed and mostly cold-based ice, we approximate that internal deformation alone sets glacier-surface velocity:

$$u(H) = u_{d,H} = \frac{2A_c}{n+1} \tau_b^n H. \tag{3}$$

Here, $\tau_b$ is the basal shear stress, $A_c$ is the Arrhenius creep constant, and $n$ is Glen's flow exponent. Our use of the basal shear stress instead of the full driving stress for glacier motion comes from the shallow-ice approximation. Through this, we assume that local stresses induced by the ice are much greater than stresses induced lateral coupling between columns of ice. We next





expand basal shear stress, $\tau_b$, into measurable parameters:

$$\tau_b = f\rho_i g H \sin(\alpha). \tag{4}$$

Here, $f$ is a shape factor accounting for lateral drag along glacier margins (Gantayat et al., 2014; Jiskoot, 2011; Linsbauer et al., 2012), $\rho_i$ is the density of ice, $g$ is gravitational acceleration, $\alpha$ is ice-surface slope angle, and $H$ is ice thickness.

Combining equations 3 and 4 and rearranging them to solve for ice thickness, $H$, gives the following expression:

$$H = \left(\frac{n+1}{2A_c(f\rho_i g)^n}\right)^{1/(n+1)}\left(\frac{u(H)}{\sin(\alpha)^n}\right)^{1/(n+1)}. \tag{5}$$

Here, the first bracket contains defined parameters and the second bracket contains observations obtained from GIV ($u(H)$)
and a digital elevation model ($\sin(\alpha)$). We use parameter values from Cuffey and Paterson (2010); Gantayat et al. (2014) and Haeberli and Hoelzle (1995): $A_c = 3.24 \times 10^{-24}$ Pa$^{-3}$s$^{-1}$, $n$=3, $f$=0.9, $\rho_i = 917$ kg.m$^{-3}$ and $g = 9.79$ m.s$^{-2}$. We compute ice-surface slope, $\alpha$, from elevation data collected by the Shuttle Radar Topography Mission (SRTM GL1: Farr et al., 2007). The longitudinal coupling length is around one to three ice thicknesses in valley glaciers (Kamb and Echelmeyer, 1986), which is on the order of 150 m at Chimborazo. To ensure that we provide results that are consistent with the shallow-ice
approximation, we resample ice-surface slope derived from the Shuttle Radar Topography Mission data to a 150 m resolution average. Thus, the only unknown required to solve for ice thickness, $H$, is ice-surface velocity.

Chimborazo poses challenges to feature-tracking-based ice velocimetry, as its glaciers are small, often feature-poor or snow-covered, very slow moving, and regularly cloud covered. The velocity limitations are mitigated by using only images with large temporal separation (acquisition dates more than six months apart). GIV is also well suited for extracting velocities from
partially clouded imagery. We run GIV on the full Sentinel-2 dataset, which comprises 3090 image pairs with separation of more than six months. The runtime for this calculation is approximately 2 hours on a Dell XPS 15 laptop (2×16GB DDR4 2666 MHz memory, 6-core Intel i7-8750H 2.20 GHz processor). We then use Equation 5, above, to compute a spatially distributed ice-thickness map for Chimborazo.

Figures 10 and 11 show the initial ice-velocity product and the final ice-thickness map (inversion code available online from
Van Wyk de Vries, 2020c). Ice is thickest on the eastern outlet glaciers, consistent with higher precipitation on the eastern flank of Chimborazo (Saberi et al., 2019). Ice is also thicker on the flat summit plateau where a 54.4 m ice core was drilled to bedrock in 2000 (Ramirez et al., 2003; Schotterer et al., 2003), within 10% of the 60 m ice thickness the model predicts in this location. If this velocity-based inversion is correct, it implies that the Chimborazo ice cap is thickening despite rising temperatures at an average rate of 0.11 °C per decade since the 1980s (La Frenierre and Mark, 2017), or around 0.2 °C since the summit ice core
was drilled to bedrock. Ice-cap thickening could be explained by an increase in accumulation, as has been proposed for other South American glaciers (e.g. Warren et al., 1997). A higher-than-expected modelled ice thickness could also be explained by uncertainties in the inversion parameters or by invalidities in the assumption of purely internal-deformation-driven surface velocity. A lower bulk ice density ($\rho_i$), Arrhenius creep constant ($A_c$) or Glen's flow law exponent ($n$) would also all lead to lower estimates of ice thickness.





Integrating the ice thickness over the entire area shows that Chimborazo's ice cap stores around $3.9 \times 10^8$ m$^3$ of ice, or just over a third of a cubic kilometre. Most of this ice volume is stored within the summit plateau and east-verging outlet glaciers, with little ice remaining in south- and west-facing outlet glaciers. This is less than half of Farinotti et al. (2019)'s estimate for the ice volume at Chimborazo in the year 2000 ($8.3 \times 10^8$m$^3$). Part of this difference is related to the larger glacier extent polygon from the year 2000 used by Farinotti et al. (2019) in their calculation. This ice polygon may incorrectly include snow covered

bedrock, and represent an over-estimation of the ice extent in the year 2000 (e.g. La Frenierre and Mark, 2017). Farinotti et al. (2019) do not use any ice-velocity data, but use several inversion methods based on glacier geometry, topography, climate and ice thickness measurements. Figure 11 compares the ice-thickness map created by inverting GIV ice velocities and from Farinotti et al. (2019). They predict that ice was thickest (>100 m thick) on the western flank of Chimborazo. The majority of this area is now entirely ice free, which would require ice thinning of up to 5 m per year to reconcile with Farinotti et al.

(2019)'s computed ice thicknesses.

## 4   Discussion

These four examples underline the versatility of GIV for calculating ice velocities in diverse environments. GIV's usefulness derives from its flexibility, ease of use, and ability to rapidly process large datasets. Most regular laptop and desktop computers now include at least 4 cores, which GIV uses to speed up calculations by a factor of two or more (Figure 3). This makes

velocity-field calculations with hundreds to thousands of image pairs possible on regular computers. The inclusion of 'temporal oversampling' allows much larger datasets to be generated than via simple consecutive-image comparison; a dataset of 100 images may in fact include several thousand usable image pairs. We combine methodological advances in feature tracking and image processing from both geoscience and engineering toolboxes, and develop new filtering techniques to improve the quality of the final surface-velocity maps. GIV provides a rapid and easy to use interface (shown in Figure 12) and a user manual, and

may also be of use to communities who would not generally be involved with glacier remote sensing (Van Wyk de Vries, 2020a, b).

    Other feature-tracking algorithms used in glaciological research include CARST (Cryosphere and Remote Sensing Toolkit: Willis et al., 2018), COSI-corr (Co-registration of Optically Sensed Images and Correlation: Leprince et al., 2007b), AutoRIFT (Autonomous Repeat Image Feature Tracking: Gardner et al., 2018), and SenDiT (the Sentinel-2 Displacement Toolbox: Nagy

and Andreassen, 2019; Nagy et al., 2019). CARST contains a mixture of Python and Bash scripts used to monitor changes in glaciers, and includes feature-tracking and ice-elevation-change-monitoring tools (Willis et al., 2018; Zheng et al., 2019a, 2018). COSI-Corr is a flexible co-registration and feature-tracking tool written in IDL, implemented in the ENVI GIS package, and initially used for measuring co-seismic deformation. Auto-RIFT is a Python-based feature-tracking algorithm (Gardner et al., 2018) with similar core components to GIV. It was used to calculate yearly resolution average velocity maps of the

Antarctic and Greenland ice sheets (ITS_LIVE dataset). SenDiT provides a platform to automatically download and generate velocity maps based on Sentinel-2 data, using a Python interface with bindings to the C and Fortran based *imcorr* toolbox (Scambos et al., 1992) for feature-tracking calculations.

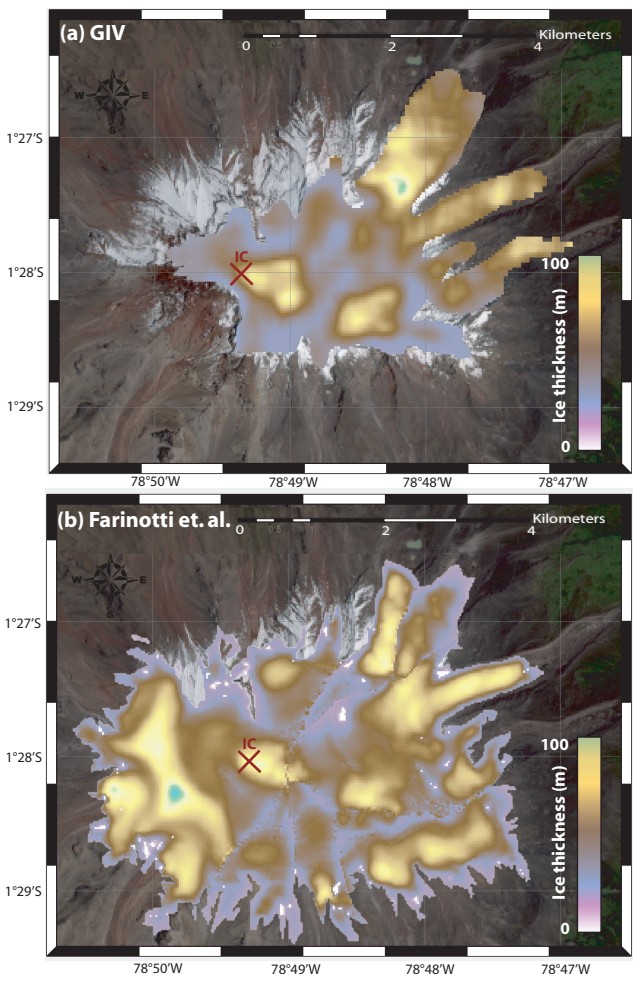

**Figure 11.** Ice thickness for the Chimborazo ice cap, Ecuador inverted from GIV (a) and from Farinotti et al. (2019) (b). The cross labeled 'IC' marks the position an ice core was drilled at Cumbre Ventimilla in 2000, reaching bedrock at 54.4 m (Schotterer et al., 2003). Note Farinotti et al. (2019) estimate ice thicknesses of >100 m in areas on the western flank that are now ice free. Imagery from Sentinel-2.

In some circumstances, GIV will not be the most suitable feature tracking tool. For example, users requiring prior co-registration of images (e.g. with airphotos) may still wish to use COSI-Corr. The objective of GIV is not to compete with or replace all the above tools, but rather to provide an easy to use, flexible and robust alternative. GIV is quick to learn and fast to run, and results derived with it are easy to reproduce. GIV allows users to modify image-processing and feature-tracking parameters based on their expert knowledge of particular glaciers, without the need for specific computational knowledge. GIV may be run either directly through MATLAB functions, through a MATLAB graphical user interface (Van Wyk de Vries, 2020a), or as an independent desktop app that may be run with no MATLAB license (Van Wyk de Vries, 2020b). GIV has been tested and successfully run on Windows, Mac and Linux operating systems.



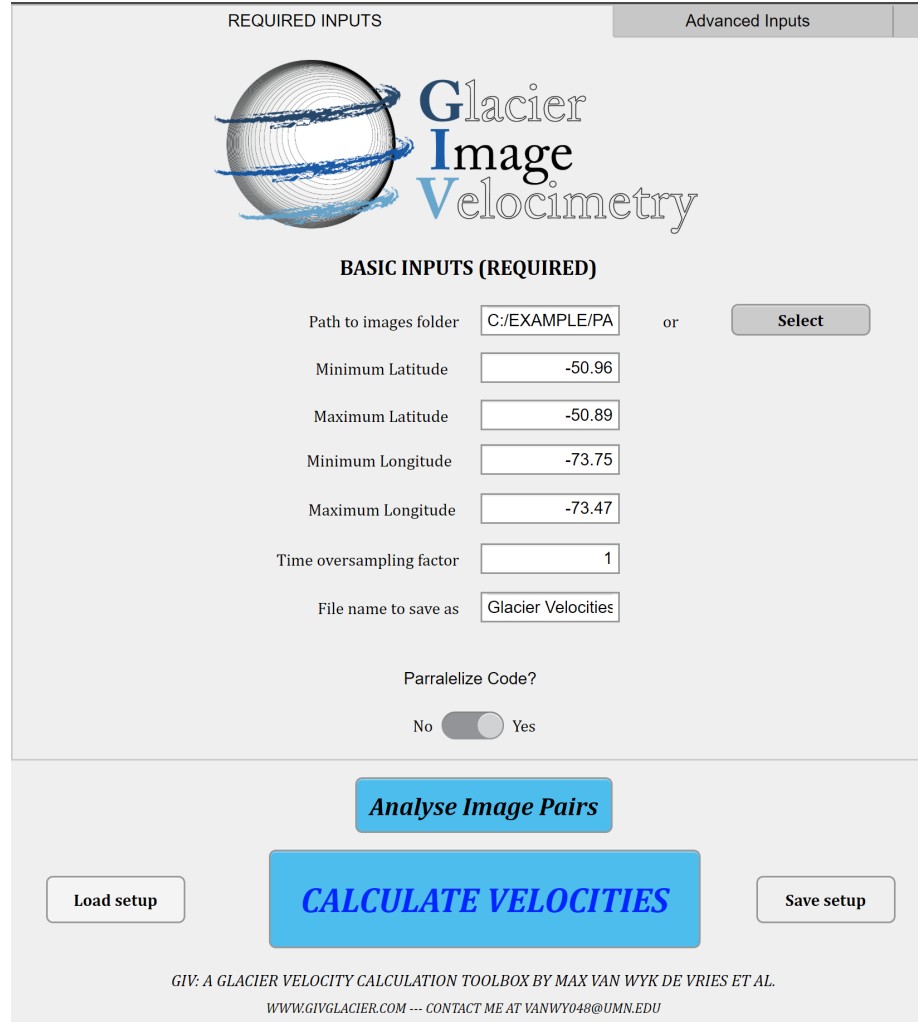

**Figure 12.** Main graphical user interface for GIV, showing the main input fields. This interface may either be run through MATLAB or as an independent desktop app with no licensing requirements.

## 5 Conclusions

GIV is a versatile, GUI-based, and fully parallelised toolbox that enables rapid calculation of glacier velocity fields from satellite imagery. GIV incorporates recent improvements in optical satellite imagery availability and resolution to extract high temporal and spatial resolution velocity maps, and uses novel and pre-existing filters to optimise the quality of these velocity maps. GIV has been successfully tested on a wide range of environments, including small valley glaciers (Glacier d'Argentière, France), tropical ice caps (Volcán Chimborazo, Ecuador), and large outlet glaciers (Glaciar Perito Moreno, Argentina, and outflow from the Vavilov Ice Cap, Russia). We show that ice-velocity datasets are versatile and may be used to compliment




field campaigns, study glacier dynamics, and make ice-volume estimates. Source code and pre-compiled binary executables for GIV are availble from Van Wyk de Vries (2020a) and Van Wyk de Vries (2020b).

*Code availability.* MATLAB code for GIV may be downloaded from https://github.com/MaxVWDV/glacier-image-velocimetry (Van Wyk de Vries, 2020a). The GIV standalone app may be downloaded from https://github.com/MaxVWDV/glacier-image-velocimetry-app (Van Wyk de Vries, 2020b). Both include a user manual and examples. Code to invert ice velocity for ice thickness may be downloaded from https: //github.com/MaxVWDV/glacier-velocity-to-thickness-inversion (Van Wyk de Vries, 2020c).

*Author contributions.* MV and AW planned the project. MV wrote the code and ran the examples. MV and AW wrote and edited the
manuscript.

*Competing interests.* The authors declare no competing interests.

*Acknowledgements.* MV was supported by a University of Minnesota College of Science and Engineering fellowship. Ben Popken assisted with early testing of GIV. Emi Ito, Kelly MacGregor, Jeff La Frenierre, Matias Romero, Shanti B. Penprase, Jabari Jones and Kerry L. Callaghan provided comments on this manuscript. This material is based upon work supported by the National Science Foundation under
Grant No. EAR-1714614, coordinated by Lead PI Maria Beatrice Magnani.



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
