# Peer review of "Glacier Image Velocimetry: an open-source toolbox for easy and rapid calculation of high-resolution glacier-velocity fields"

_The Cryosphere, 2020_

## Referee Comment (RC1) · Ted Scambos (Referee) · 20 Sep 2020

The study describes a new ice-velocity mapping toolkit using visible – near-infrared image pairs or multiple images spanning a range of time. The authors have applied several well-used and a few clever filtering and information-extraction methods in the toolkit. It is good to have one software package that provides both the vectors and a thorough means of editing them in one workflow. The authors then demonstrate the value of the velocity mapping with a group of case studies spanning the range of glacier and small ice cap environments in the northern hemisphere and a tropical location.

[Figure]

This is a well-written paper, and the method seems sound and very useful, although there are several similar tools available at this time. This should be published with minor revisions. The only major change I suggest is removing the ice thickness estimation and place it in another paper with other similar targets so that the calculations will be more visible to the community. It is not necessary to place it in this method-and-validation paper. I make several significant suggestions for the abstract as well, and many further suggestions in the rest of the text.

In general, references should be listed in time order, from earliest publication date to most recent. Adopting this convention will mean several minor changes in the manuscript.

Suggested changes to Abstract: We present 'Glacier Image Velocimetry' (GIV), an open-source and easy-to-use software toolkit for rapidly calculating high spatial resolution glacier-velocity fields. Glacier ice velocity fields reveal their flow dynamics, ice flux stability, and (with additional data and modelling) ice thickness. Obtaining glacier-velocity measurements over wide areas with field techniques is labour intensive, and often a safety risk. Recent increased availability of high-resolution, short-repeat-time optical imagery allow us to obtain ice displacement fields using 'feature tracking' based on the presence of persistent irregularities on the ice surface, and hence, velocity over time. GIV is fully parallelized, and automatically detects, filters, and extracts velocities from large datasets of images. Through this coupled toolchain and an easy-to-use GUI, GIV can rapidly analyze hundreds to thousands of image pairs, requiring only a moderately high-end laptop or desktop computer. We present four examples of how the GIV toolkit may be used: to complement a glaciology field campaign (Glaciar Perito Moreno, Argentina), calculate the velocity fields of small (Glacier d'Argentière, France) and very large (Vavilov ice cap, Russia) glaciers, and determine the ice volume present within a tropical ice cap (Volcán Chimborazo, Ecuador). Fully commented code and a standalone app for GIV are available from GitHub and Zenodo.

Consider adding these very pertinent additional references in the introduction Line 20-

21 : Howat, I.M., Porter, C., Smith, B.E., Noh, M.J. and Morin, P., 2019. The Reference Elevation Model of Antarctica. Cryosphere, 13(2), https://doi.org/10.5194/tc-13-665-2019 Scambos, T.A., Haran, T.M., Fahnestock, M.A., Painter, T.H. and Bohlander, J., 2007. MODIS-based Mosaic of Antarctica (MOA) data sets: Continent-wide surface morphology and snow grain size. Remt. Sens. Env., 111(2-3), 242-257, https://doi.org/10.1016/j.rse.2006.12.020. Line 32: Stearns, L.A., Smith, B.E. and Hamilton, G.S., 2008. Increased flow speed on a large East Antarctic outlet glacier caused by subglacial floods. Nature Geoscience, 1(12), 827-831, https://doi.org/10.1038/ngeo356. Line 42: Bindschadler, R.A. and Scambos, T.A., 1991. Satellite-image-derived velocity field of an Antarctic ice stream. Science, 252(5003), 242-246, https://doi.org/10.1126/science.252.5003.242. Line 47: Fahnestock, M., Scambos, T., Moon, T., Gardner, A., Haran, T. and Klinger, M., 2016. Rapid large-area mapping of ice flow using Landsat 8. Remt. Sens. Env., 185, 84-94, https://doi.org/10.1016/j.rse.2015.11.023. Line 52: you may want to note these two data sites, presenting already-processed data – https://nsidc.org/data/golive https://nsidc.org/apps/itslive/ Table 1: PyCorr is the tool behind Fahnestock et al., 2016, which produced some of the mosaics in Gardner et al., 2018.

Line 117 – you say 'multipass methods take advantage of the reduction in chip size to improve the signal to noise'. I think this needs to be rephrased – in general, if there is low shear or deformation across the scene, large chip sizes produce much better matches.

Line 150 – at what 'scale' or number of grid cells are these statistical values calculated? I would assume this scale is either set by the user or by some extracted geography of the ice within the image pair(s).

Figure 5 – label the color bars, with 'Flow Speed' and 'Bearing'…. Could also add degree symbols to the bearing indices,

Figure 7 – the perspective view is a bit difficult to follow without somewhat more area

covered to gain a feel for the 3-dimensional structure. . ... The figure is nice but takes a while to orient mentally. Expand view, or, a second inset that shows the map view?

Figure 8 – Expand the velocity scale (taller) in one of the top two insets, and no need to repeat it in both (a) and (b). The titles of (a) and (b) should be 'ice speed' unless you include a few vectors for direction. Include the month of the velocity mapping in the 'title' of the insets for (a) and (b).

Line 263 – suggest change to '. ...or ice basal conditions are identified.'

Figure 9 – What is the difference, exactly? GIV minus Zheng or Zheng minus GIV?. The scale of the speed differences is large for the margins, and appears to be locally consistent. However it does not extend outside of the glacier boundaries, so it would seem that its not due to a rotational mis-registration. It would seem that somehow the two mappings captured a true physical shift in the margins or speed profile across Vavilov during May 2017 somehow.

Lines 268-269 – Was geolocation necessary? Landat 8 geolocation is generally within 5 meters, i.e. significantly less than a panchromatic band pixel; Sentinel-2 image geolocation is similar. What was the scale of the error in geolocation that you corrected?

Line 272 – remove 'total', makes it sound like you are summing the velocities. . .note also, it's speed, not velocity: velocity is a vector.

Line 290 – please provide the lat long for Chimborazo, and for the other sites (sorry if I missed it).

Line 295 – change to '. . .but for this single point (which we use for benchmarking our method), combined with. . ."

Lines 306 – 316: Hmm. Do you not have a thermal profile from the 54 meter core to the base? It is very likely that the base of the ice is warmer than the air-temperature-based isotherm because of insulation. Moreover, the presence of waterlogged ice (a firn aquifer) means that it is likely that water drains to the bed – and further, you note that

the glacier supplies water to the local watershed so seasonal melting is significant (and in general, meltwater on a glacier finds its way through the ice and to the ice-bedrock interface, warming the bedrock.

Rather than take on this complex mountain glacier, why not apply your method to the Vavilov Ice Cap outside of the area of sudden rapid flow? Or you might try a glacier where it is more certain that <0°C conditions exist at the bed, and with more validation data – Commonwealth Glacier or Canada Glacier in the Dry Valleys would be good.

More generally — this paper does not need this section on thickness estimation – your point is to show off the quality and extent and usefulness of the GIV data, and the extensive processing and filtering steps you take – and while this is a demonstration of 'usefulness', it's better as a stand-alone study of Chamborazo or a small set of glaciers where the result will not be lost in the literature (no one will find your thickness estimate in this paper). A series of ice thickness estimates for cold-based Andean glaciers, or Dry Valley glaciers, or selected Himalayan glaciers, using GIV, would be cited extensively.

Line 322 – re-write, confusing. Maybe you mean: local -basal- stresses induced by the ice are much greater than lateral stresses between columns of ice. . .?

Line 340 – 3090 Sentinel -2 image pairs in 2 hours on a Dell laptop – did you subset the Sentinel-2 to just cover the Chimborazo summit area? I am also surprised there are that many pairs – you might include a statement as to how many distinct images were processed.

Lines 344-355 – There is no need for all this speculation on your ice thickness estimate in this paper – this belongs in a separate paper where the approach can be developed more and applied to a set of related areas (perhaps). I strongly suggest cutting this ice thickness section out and placing it in a separate paper. I think it might be interesting to apply GIV to an ice sheet region such as Nimrod Glacier or Peterman Glacier.

Line 390 – change to '...alternative. GIV is easily learned and is not computationally time-consuming, and the results....' Not to be harsh, but GIV itself does not learn, and doesn't run either.

---

## Referee Comment (RC2) · Anonymous Referee #2 · 1 Oct 2020

The toolbox presented by the authors is a GUI to ease feature tracking routines within a MATLAB environment. The naming of the toolbox is a reference to Particle Image Velocimetry (PIV), which is a well established technique within fluid mechanics. It is based upon particle seeding within a fluid, illuminated by light behind a narrow slit to get a within-plane velocity profile, thus in a well-controlled environment. When particles are not well stirred, have a lot of out-of-plane displacements, too large time separation, or other effects complicating the tracking: the experiment is done all over. The authors lend the methodologies from this domain, and implement this on natural environment

[Figure]

**TCD**

to extract glacier surface velocity. This transfer is more complicated as it might initially seem to be, as seeding is absent (though people in hydrology are experimenting on this issue [Pizarro et al. 2020]), and more importantly experiments can't be redone.

The toolbox has some application specific adjustments, which is in line with other application domains, where similar toolbox are introduced, such as: Optical tracking velocimetry (OTV) [Tauro et al 2018], Surface Structure Image Velocimetry (SSIV) [Leitaoa et al. 2018], Kanade–Lucas Tomasi Image Velocimetry (KLT-IV) [Perks 2020], Part2Track [Janke et al. 2020], TecPIV [Boutelier 2016], Rectification of Image Velocity Results (RIVeR) [Patalano et al. 2017], Waves Acquisition Stereo System (WASS) [Bergamasco et al. 2017], or more specific to glaciers: Environmental Motion Tracking (EMT) [Schwalbe et al. 2017], PyTrx [How et al. 2020], Image GeoRectification and feature tracking toolbox (IMGRAFT) [Messerli et al. 2015], Pointcatcher [James et al. 2016]. Given the list above, and the journals where these toolboxes are presented, it might be a valid question why the authors have opted for a submission to The Crysophere, and not a technical EGU journal like Geoscientific Instruments or Geoscientific Model Development?

There are some implementations which one could expect for a toolbox tailored towards glaciology to be presented, but are not included. In addition some features are included, which need further assessment, to highlight their improvement, if they do.

To be more specific the comments are grouped into different paragraphs: Analysis of the 10-bit data of Landsat 8 [Jeong et al. 2015] and Sentinel-2 [Kääb et al. 2016] over dark overshadowed terrain have shown there is a significant benefit over 8bit data. Why do the authors downgrade their imagery to 8-bit RGB? In the same line, the use of non-georeferenced ".jpg" or ".png" data is strange. MATLAB supports mapping tools for such issues, why are these not adopted. While the toolbox needs a specific dimension in order to work, it is very strange this is not present in the toolbox. The choice of using angles (lat,lon) for metric data is confusing.

[Figure]

The authors present a new image transform dubbed "NAOF", but is there a significant improvement, over for example typical orientation filtering approach? At least to me, the 45 degrees filters seem like a redundant feature, as 90deg angled steerable filters incorporate all information [Freeman et al. 1991]. It seems the implementation is a combination of work similar to [Ahn et al. 2011], and orientation filtering [Heid et al. 2012]. But it is questionable, if the additional calculations add towards improvement. The filter banks are correlated (as just mentioned), furthermore, the visible bands are similiarly correlated over glaciers. Hence, the information gain might be very limited.

The paragraph on velocity calculations talks about two methods to extract displacements from an image pair. However, this distinction is more about the option if refinement is applied or not. The authors limit themselves to frequency domain methods, which is the de-facto procedure for PIV. However, spatial domain methods can be more favorable for glacier related applications. This comes down to the point given above, of the transfer from fluid mechanics towards an application domain without the ability to redo an experiment. Hence, the velocity profile can be highly variable (extensive shear, valley walls, diverging flow, fast flow, ...). Spatial domain methods are more resistant to shear flow, the chip is not related to the search space (as with fourier methods). The authors might have considered spatial domain methods, but for clarity it might be fruitfull to mention this, and why. In PIV experiments, one is able to adjust the image sampling rate, to be in accordance with the maximum flow velocity, this is less the case for glaciers.

Technical comments: In general the code is very messy, many duplicate lines exist, commenting is absent ("strcmpi(inputs{51,2}, 'Yes')"). At multiple locations indents are used in-appropriately. Documentation and commenting within the code is limited. Every function has an extensive help section, which is typically an acknowledgment and info about GIV. Input or output variables are not described. For many or all functions, tests are lacking. Factorization is on a low level. Given this situation of the code, how do the authors see adoption by others? The objective of the authors is to be a wrapper (based upon ImGraft and matPIV), then one would expect documentation to be extensive.

Sub-pixel: There does not seem to be a sub-pixel estimation present? Why is this?

The domain of PIV is also extensively populated with papers about peak-locking, it would be good if the authors put some effort in this as well. This bias is especially present in sub-pixel estimation directly done on the correlation surface. Other methods, such as TPSS, as implemented in Cosi-Corr are less sensitive to this effect (which to a large extent might explain its popularity in geodetic imaging).

Geo-referencing: The gridspacing as it is implemented now assumes all rectangular and upright pixels, please adjust this.

Matlab: To my knowledge Matlab is not open access, the code might be open, but many of the algorithms are hidden away and licensing is needed. Hence, I am not so sure if the description about FFTW is needed, as one is not able to access this. Secondly, it is a standard routine within MATLAB.

Merging satellite data: The authors describe a time-series construction method, illustrate a toy example. But it seems they apply a sigma-filter, and do simple infilling. There are more advanced methods available, which are more adaptive towards velocity data. Hence, a simple reference to (e.g.: [Mouginot et al. 2017]) might suffice for the implementation here, and this section can thus be reduced considerably.

The temporal sampling is a bit counter intuitive, as this is an opposite direction to the workflow of [Millan et al. 2019]. Where they found the time-span needs to be of sufficient size, in order to be of most use. Why then do the authors prefer the shorter time steps?

Code specific: There is a function about intensity capping, while this might be of interest to PIV, it is questionable if this is of interest to environmental signals. Bright intensity of seed points within dark water are very much different from glacier surfaces. Also how

does this merge with the high-pass filtering... ?

Even though your algorithms are optimized, they might not pull out the best of the machinery, yet. For example, the function "neighbourfilter" contains a double loop, to process a kernel. An internal function of Matlab called "nlfilter" might be much faster. A good reference for such implementations can be found in "Accelerating MATLAB Performance" (http://undocumentedmatlab.com/books/matlab-performance). In the same function from line 95 onward, the authors use double indexing. While linear indexing is possible as well, which make vectorization possible (which is MATLABs strong point). This greatly improves the processing time as well.

I am not sure if the variance calculation of the flow direction is calculated correctly. Offsetting the direction and applying a weighting might work, but maybe correct circular statistics might be more appropriate, see [Berens 2009] for a toolbox implementation. It might be more easy to use the mapping coordinates instead...?

It might be good to give an example; here is a sniplet of your code (a straight copy):

if smoothsize == 2;

mask = [0 1 0; 1 0 1; 0 1 0 ];

elseif smoothsize == 3;

mask = [0 1 0;1 1 1; 1 0 1; 1 1 1; 0 1 0 ];

elseif smoothsize == 4;

mask = [0 0 1 0 0 ;0 1 1 1 0 ;0 1 1 1 0; 1 1 1 1 1 ; 0 1 1 1 0; 0 1 1 1 0; 0 0 1 0 0 ];

elseif smoothsize == 5;

mask = [0 0 1 0 0 ;0 1 1 1 0 ;0 1 1 1 0;1 1 1 1 1 ;1 1 1 1 1 ; 1 1 1 1 1 ; 0 1 1 0; 0 1 1 1 0; 0 0 1 0 0 ];

end

why isn't this in a function? Something like:

```
function [mask] = make_mask(smoothsize)
"

generates a neighborhood kernel in the form of a diamond,

though excludes the central pixel

input:

smoothsize - integer value

output:

mask - logical array

"

if nargin<1, smoothsize = 3; end

mask_radius = floor(smoothsize/2);

mask = strel('diamond', mask_radius); % make a diamond shape

mask = mask.Neighborhood;

mask(mask_radius+1, mask_radius+1) = 0; % exclude the central element

end
```

In all I am aware this is, like all research, work in progress, and I think this is a very useful direction. But in its current state and form, sufficient work needs to be done to be of interest, please see [Perks 2020] for a nice example. Here workflows are nicely presented, a dashboard is present, etc. All in all, I think a transfer towards a technical journal of EGU might be more in place.

Textual: p63: "broken down" is misleading, as overlapping chips can also be used

p140: superscript the 2, to make it a square

references:

Ahn et al. 2011 10.1109/TGRS.2011.2114891

Berens 2009 10.18637/jss.v031.i10

Bergamasco et al. 2017 10.1016/j.cageo.2017.07.001

Boutelier 2016 10.1016/j.cageo.2016.02.002

Freeman et al. 1991 10.1109/34.93808

James et al. 2016 10.1017/jog.2016.27

Janke et al. 2020 10.1016/j.softx.2020.100413

Jeong et al. 2015 10.1016/j.rse.2015.08.023

Heid et al. 2012 10.1016/j.rse.2011.11.024

How et al. 2020 10.3389/feart.2020.00021

Kääb et al. 2016 10.3390/rs8070598

Leitaoa et al. 2018 10.1016/j.jhydrol.2018.09.001

Messerli et al. 2015 10.5194/gi-4-23-2015

Millan et al. 2019 10.3390/rs11212498

Mouginot et al. 10.3390/rs9040364

Tauro et al. 2018 10.3390/rs10122010

Patalano et al. 2017 10.1016/j.cageo.2017.07.009

Perks 2020 10.5194/gmd-2020-187

Pizarro et al. 2020 10.5194/hess-2020-188

Schwalbe et al. 2017 10.5194/esurf-5-861-2017

---

## Author Comment (AC1) · 24 Oct 2020

**Glacier Image Velocimetry: an open-source toolbox for easy and rapid calculation of high-resolution glacier-velocity fields**

Maximillian Van Wyk de Vries[1,2] and Andrew D. Wickert[1,2]

[1]Department of Earth & Environmental Sciences, University of Minnesota, Minneapolis, MN
[2]Saint Anthony Falls Laboratory, University of Minnesota, Minneapolis, MN

**Correspondence:** Maximillian Van Wyk de Vries (vanwy048@umn.edu)

**Final response - Review 1**

Editor comments are given in *italic*, responses in regular font.

5     *The study describes a new ice-velocity mapping toolkit using visible – near-infrared image pairs or multiple images spanning a range of time. The authors have applied several well-used and a few clever filtering and information-extraction methods in the toolkit. It is good to have one software package that provides both the vectors and a thorough means of editing them in one workflow. The authors then demonstrate the value of the velocity mapping with a group of case studies spanning the range of glacier and small ice cap environments in the northern hemisphere and a tropical location.*

10     *This is a well-written paper, and the method seems sound and very useful, although there are several similar tools available at this time. This should be published with minor revisions. The only major change I suggest is removing the ice thickness estimation and place it in another paper with other similar targets so that the calculations will be more visible to the community. It is not necessary to place it in this method-and-validation paper. I make several significant suggestions for the abstract as well, and many further suggestions in the rest of the text.*

15

We thank Dr. Ted Scambos for the positive comments about the method and manuscript, as well as the useful suggestions. We have made several key changes to the manuscript based on the recommendations in this review, in particular:

1. We added a number of key references.

2. We rewrote of the abstract based on your suggestions.

20    3. We removed the section on ice thickness inversion, which will be developed into its own manuscript (see detailed comments below)

4. We edited our figures for clarity.

*In general, references should be listed in time order, from earliest publication date to most recent. Adopting this convention will mean several minor changes in the manuscript.*

We have verified that all references are now listed in chronological order.

*Suggested changes to Abstract: We present 'Glacier Image Velocimetry' (GIV), an open-source and easy-to-use software toolkit for rapidly calculating high spatial resolution glacier-velocity fields. Glacier ice velocity fields reveal their flow dynamics, ice flux stability, and (with additional data and modelling) ice thickness. Obtaining glacier velocity measurements over wide areas with field techniques is labour intensive, and often a safety risk. Recent increased availability of high-resolution, short-repeat-time optical imagery allow us to obtain ice displacement fields using 'feature tracking' based on the presence of persistent irregularities on the ice surface, and hence, velocity over time. GIV is fully parallelized, and automatically detects, filters, and extracts velocities from large datasets of images. Through this coupled toolchain and an easy-to-use GUI, GIV can rapidly analyze hundreds to thousands of image pairs, requiring only a moderately high-end laptop or desktop computer. We present four examples of how the GIV toolkit may be used: to complement a glaciology field campaign (Glaciar Perito Moreno, Argentina), calculate the velocity fields of small (Glacier d'Argentière, France) and very large (Vavilov ice cap, Russia) glaciers, and determine the ice volume present within a tropical ice cap (Volcán Chimborazo, Ecuador). Fully commented code and a standalone app for GIV are available from GitHub and Zenodo.*

We are very grateful for this re-write of our abstract, and adopt it with a few minor changes (in particular to reflect the removal of the ice thickness/volume calculations).

*Consider adding these very pertinent additional references in the introduction Line 20-21 : Howat, I.M., Porter, C., Smith, B.E., Noh, M.J. and Morin, P., 2019. The Reference Elevation Model of Antarctica. Cryosphere, 13(2), https://doi.org/10.5194/tc-13-665-2019 Scambos, T.A., Haran, T.M., Fahnestock, M.A., Painter, T.H. and Bohlander, J., 2007. MODIS-based Mosaic of Antarctica (MOA) data sets: Continent wide surface morphology and snow grain size. Remt. Sens. Env., 111(2-3),242-257, https://doi.org/10.1016/j.rse.2006.12.020. Line 32: Stearns, L.A., Smith, B.E. and Hamilton, G.S., 2008. Increased flow speed on a large East Antarctic outlet glacier caused by subglacial floods. Nature Geoscience, 1(12), 827-831, ://doi.org/10.1038-/ngeo356. Line 42: Bindschadler, R.A. and Scambos, T.A., 1991. Satellite-image-derived velocity field of an Antarctic ice stream. Science, 252(5003), 242-246, https://doi.org/10.1126/science.252.5003.242. Line 47: Fahnestock, M., Scambos, T., Moon, T., Gardner, A., Haran, T. and Klinger, M., 2016. Rapid large-area mapping of ice flow using Landsat 8. Remt. Sens. Env., 185, 84-94, https://doi.org/10.1016/j.rse.2015.11.023.*

We agree that these references provide important background information, and have added them to our introduction.

*Line 52: you may want to note these two data sites, presenting already-processed data – https://nsidc.org/data/golive https://nsidc.org/apps/itslive/*

60     We have added a one sentence description of these datasets, as they are likely of interest to the readers of this paper (and may be a viable substitute for running GIV over large regions, and where lower spatial and temporal resolution are acceptable).

*Table 1: PyCorr is the tool behind Fahnestock et al., 2016, which produced some of the mosaics in Gardner et al., 2018.*

65     We have add PyCorr to the table, along with 3 other tools that were flagged by reviewer 2 (Pointcatcher, PyTrx and EMT). We have also added a sentence to the table description to highlight that our list is not exhaustive. We were unable to locate a copy of PyCorr software on the internet (although it is mentioned in several papers, as you have highlighted), and are unsure if it is available for use by other teams.

70    *Line 117 – you say 'multipass methods take advantage of the reduction in chip size to improve the signal to noise'. I think this needs to be rephrased – in general, if there is low shear or deformation across the scene, large chip sizes produce much better matches.*

    We have expanded and clarified the benefits of multipass methods in the following sentences: "Multi-pass methods refine
75 displacement estimates in multiple iterations, refining initial coarse window size displacement calculations with progressively smaller window sizes. Multi-pass methods combine the advantages of better feature matching at large window sizes with the higher spatial resolution of small window sizes. Both methods are integrated into GIV, which uses a 3 iteration multi-pass algorithm."
    We have also added a reference in the methods section to a recent PhD thesis by Dr. Bas Altena (2018), which provides a
80 well written and detailed background on some of the common processing steps in glacier feature tracking.

   *Line 150 – at what 'scale' or number of grid cells are these statistical values calculated? I would assume this scale is either set by the user or by some extracted geography of the ice within the image pair(s).*

85     The statistical values are calculated for a single cell, averaged through time. For clarity, the sentence was changed to : "Secondly, GIV calculates the mean, standard deviation, median, minimum, and maximum velocities through time at each grid cell in the dataset."

   *Figure 5 – label the color bars, with 'Flow Speed' and 'Bearing... Could also add degree symbols to the bearing indices,*
90

    The recommended changes have been made.

   *Figure 7 – the perspective view is a bit difficult to follow without somewhat more area covered to gain a feel for the 3-dimensional structure... The figure is nice but takes a while to orient mentally. Expand view, or, a second inset that shows the*

 *map view?*

This figure has been edited to include a second inset showing the full-glacier velocities in map view. The color scheme has also been changed (from https://colorbrewer2.org/) in order to be 1) suitable for converting to black and white and 2) color-blind friendly, as recommended by the editor.

*Figure 8 – Expand the velocity scale (taller) in one of the top two insets, and no need to repeat it in both (a) and (b). The titles of (a) and (b) should be 'ice speed' unless you include a few vectors for direction. Include the month of the velocity mapping in the 'title' of the insets for (a) and (b).*

 The velocity scale has been expanded, labelled as 'Flow speed (m/yr)', and is no longer duplicated. The velocity maps are full yearly averages (well, March-September due to the Arctic winter), and so do not correspond to a specific month. They have been labelled as yearly averages.

*Line 263 – suggest change to '....or ice basal conditions are identified.'*

The recommended change has been made.

*Figure 9 – What is the difference, exactly? GIV minus Zheng or Zheng minus GIV?. The scale of the speed differences is large for the margins, and appears to be locally consistent. However it does not extend outside of the glacier boundaries, so*
 *it would seem that its not due to a rotational mis-registration. It would seem that somehow the two mappings captured a true physical shift in the margins or speed profile across Vavilov during May 2017 somehow.*

We have edited the figure caption to clarify the difference map, it now reads: "a) shows a difference map, corresponding to Zheng et al. velocity minus GIV velocity"
    We have also been in touch with the creator of the Vavilov velocity maps (Whyjay Zheng) and it indeed seems likely that the difference may be resolving a real shift in margin position and/or velocity over time. It is relevant to note that a project comparing results from GIV and other feature tracking tools (including Zheng et al's CARST) to glacier GPS data is ongoing.

*Lines 268-269 – Was geolocation necessary? Landat 8 geolocation is generally within 5 meters, i.e. significantly less than*
 *a panchromatic band pixel; Sentinel-2 image geolocation is similar. What was the scale of the error in geolocation that you corrected?*

Geolocation shift was small, on the order of half a velocity pixel to one velocity pixel (~50-100m). GIV velocities were derived from non-georeferenced imagery (.jpg) and georeferenced based on corner coordinates, which likely contributed to the

130    mis-match. An option to run GIV directly on geotiff images has since been added.

*Line 272 – remove 'total', makes it sound like you are summing the velocities... note also, it's speed, not velocity: velocity is a vector.*

135    The recommended change has been made, and 'velocity' changed to 'speed' where referring to the magnitude only.

*Line 290 – please provide the lat long for Chimborazo, and for the other sites (sorry if I missed it).*

The coordinates of all locations have been added.

140

*Line 295 – change to '...but for this single point (which we use for benchmarking our method), combined with...'*

*Lines 306 – 316: Hmm. Do you not have a thermal profile from the 54 meter core to the base? It is very likely that the base of the ice is warmer than the air-temperature-based isotherm because of insulation. Moreover, the presence of water-*
145    *logged ice (a firn aquifer) means that it is likely that water drains to the bed – and further, you note that the glacier supplies water to the local watershed so seasonal melting is significant (and in general, meltwater on a glacier finds its way through the ice and to the ice-bedrock interface, warming the bedrock.*

We agree with these comments, and will bear them in mind when further developing the ice thickness inversion work. In
150    the meanwhile, the associated lines have been removed from the manuscript.

*Rather than take on this complex mountain glacier, why not apply your method to the Vavilov Ice Cap outside of the area of sudden rapid flow? Or you might try a glacier where it is more certain that <0C conditions exist at the bed, and with more validation data – Commonwealth Glacier or Canada Glacier in the Dry Valleys would be good.*

155

*More generally — this paper does not need this section on thickness estimation – your point is to show off the quality and extent and usefulness of the GIV data, and the extensive processing and filtering steps you take – and while this is a demonstration of 'usefulness', it's better as a stand-alone study of Chamborazo or a small set of glaciers where the result will not be lost in the literature (no one will find your thickness estimate in this paper). A series of ice thickness estimates for cold-based*
160    *Andean glaciers, or Dry Valley glaciers, or selected Himalayan glaciers, using GIV, would be cited extensively.*

Since the submission of this manuscript we have run variations of this methodology on 6 additional tropical ice caps in Ecuador and Colombia, including some glaciers with better ground-penetrating radar derived ice thickness constraints. Following your suggestions, we crop out the discussion about inverting for ice thickness, and will place this in a separate paper with more

space for discussing the inversion parameters and model-data comparison. We have kept the Chimborazo ice velocity results in this paper, as we feel it is a useful example of using GIV at small and slow-moving glaciers.

*Line 322 – re-write, confusing. Maybe you mean: local -basal- stresses induced by the ice are much greater than lateral stresses between columns of ice...?*

*Line 340 – 3090 Sentinel -2 image pairs in 2 hours on a Dell laptop – did you subset the Sentinel-2 to just cover the Chimborazo summit area? I am also surprised there are that many pairs – you might include a statement as to how many distinct images were processed.*

This sentence has been edited for clarity. There are 91 unique images, which are cropped to the region surrounding Chimborazo (with enough bare rock to enable the stable ground correction) and paired up into all possible pairs with >6 months separation. In theory there are $(n^2-n)/2$ possible image pairs, or 4095 total image pairs in this case (and around 1000 images pairs are excluded due to a separation of <6 months).

*Lines 344-355 – There is no need for all this speculation on your ice thickness estimate in this paper – this belongs in a separate paper where the approach can be developed more and applied to a set of related areas (perhaps). I strongly suggest cutting this ice thickness section out and placing it in a separate paper. I think it might be interesting to apply GIV to an ice sheet region such as Nimrod Glacier or Peterman Glacier.*

As mentioned above, we have limited the Chimborazo section to the calculation of ice velocities and will develop the thickness inversion methodology into a stand-alone paper. We have tested GIV on Antarctic Peninsula glaciers and portions of Thwaites glacier calving front (with no issues). Nimrod glacier appears to be an interesting case study due to its large central nunatak, and good baseline data from Stearns (2007, PhD thesis; 2011).

*Line 390 – change to '...alternative. GIV is easily learned and is not computationally time-consuming, and the results...' Not to be harsh, but GIV itself does not learn, and doesn't run either.*

We have corrected our sentence.

---

## Author Comment (AC2) · 24 Oct 2020

**Glacier Image Velocimetry: an open-source toolbox for easy and rapid calculation of high-resolution glacier-velocity fields**

Maximillian Van Wyk de Vries[1,2] and Andrew D. Wickert[1,2]

[1]Department of Earth & Environmental Sciences, University of Minnesota, Minneapolis, MN
[2]Saint Anthony Falls Laboratory, University of Minnesota, Minneapolis, MN

**Correspondence:** Maximillian Van Wyk de Vries (vanwy048@umn.edu)

**Final response - Review 2**

Editor comments are given in *italic*, responses in regular font.

*The toolbox presented by the authors is a GUI to ease feature tracking routines within a MATLAB environment. The naming of the toolbox is a reference to Particle Image Velocimetry (PIV), which is a well established technique within fluid mechanics. It is based upon particle seeding within a fluid, illuminated by light behind a narrow slit to get a within-plane velocity profile, thus in a well-controlled environment. When particles are not well stirred, have a lot of out-of-plane displacements, too large time separation, or other effects complicating the tracking: the experiment is done all over. The authors lend the methodologies from*

*this domain, and implement this on natural environment to extract glacier surface velocity. This transfer is more complicated as it might initially seem to be, as seeding is absent (though people in hydrology are experimenting on this issue [Pizarro et al. 2020]), and more importantly experiments can't be redone.*

     *The toolbox has some application specific adjustments, which is in line with other application domains, where similar toolbox are introduced, such as: Optical tracking velocimetry (OTV) [Tauro et al 2018], Surface Structure Image Velocimetry*

*(SSIV) [Leitaoa et al. 2018], Kanade–Lucas Tomasi Image Velocimetry (KLT-IV) [Perks 2020], Part2Track [Janke et al. 2020], TecPIV [Boutelier 2016], Rectification of Image Velocity Results (RIVeR) [Patalano et al. 2017], Waves Acquisition Stereo System (WASS) [Bergamasco et al. 2017], or more specific to glaciers: Environmental Motion Tracking (EMT) [Schwalbe et al. 2017], PyTrx [How et al. 2020], Image GeoRectification and feature tracking toolbox (IMGRAFT) [Messerli et al. 2015], Pointcatcher [James et al. 2016]. Given the list above, and the journals where these toolboxes are presented, it might be a*

*valid question why the authors have opted for a submission to The Crysophere, and not a technical EGU journal like Geoscientific Instruments or Geoscientific Model Development?*

We thank the reviewer for their detailed review. We would specifically like to thank the reviewer for taking the time to read through GIV's code, as their recommendations have helped improve both the manuscript and the toolbox itself. We have taken the time to review each comment and recommendation in detail, and have made some moderate to substantial changes to GIV's code. We hope that this response clarifies some of our choices, and highlights the changes made.

The main modification which have been made to the code are:

1. We review the entire code to improve the code formatting (removal of unnecessary spaces and tabs) and increase the number of comments. We do not expect a large proportion of GIV's users to be reading through the entire code, but aim to make it accessible for when they do (and make it possible for users to use our functions in their own tools).

2. We change the format of our inputs array (which stores the input parameters) from a MATLAB cell array to a MATLAB struct array. The struct array uses dot notation to make it more human-readable, and should make the code more accessible (e.g. changed from 'inputs{30,2}' to 'inputs.parralelize').

3. We implement an option to read in images directly as geotiff images and preserve the geographic information throughout processing. If users input geo-tiffs they do not need to input corner coordinates.

4. We review the calculation of circular statistics (flow direction mean and standard deviation), and edit Dr. Berens' CircStat toolbox to tolerate NaN values.

5. We re-compile the Windows standalone app with the changes, and compile additional apps for macOS and Linux.

We describe the changes in more detail below, and respond to specific review comments. We have added reference to the relevant toolboxes you have listed in this paragraph as well.

To briefly touch on the decision to publish in The Cryosphere (TC) rather than a technical journal, the primary objective of our toolbox is to be accessible to glaciologists who may not otherwise be involved in feature tracking. We hope that GIV will bridge a gap between coarser resolution global glacier surface velocity datasets (e.g. GoLIVE, ITS_LIVE) and the more localised, higher resolution data required by some studies or field campaigns. Other good feature-tracking toolboxes do exist (see table 1 for many of them), but can be challenging to use with no computational or remote sensing background. GIV is easy to install, quick to learn, and quick to run, without requiring background knowledge of how the code runs (including by students). We believe that the manuscript is well within the scope of TC, and that publishing in TC will better reach the intended audience of this paper than technical journals. In addition, we believe that the case studies presented here will be of interest to the glaciological readership of TC.

We should note that we did share your concerns at the outset, and indeed contacted the editorial staff at The Cryosphere about appropriate fit before submission. Their support for our submission was also key to this being the chosen venue for the paper. This is often a difficult choice when presenting a paper focused on both the application and its methods, and we appreciate the acknowledgment of this.

*There are some implementations which one could expect for a toolbox tailored towards glaciology to be presented, but are not included. In addition some features are included, which need further assessment, to highlight their improvement, if they do.*

We thank the reviewer for highlighting some recommended changes. We describe the changes we make to the code in this response.

*To be more specific the comments are grouped into different paragraphs: Analysis of the 10-bit data of Landsat 8 [Jeong et al. 2015] and Sentinel-2 [Kääb et al. 2016] over dark overshadowed terrain have shown there is a significant benefit over 8bit data. Why do the authors downgrade their imagery to 8-bit RGB? In the same line, the use of non-georeferenced ".jpg" or ".png" data is strange. MATLAB supports mapping tools for such issues, why are these not adopted. While the toolbox needs a*

*specific dimension in order to work, it is very strange this is not present in the toolbox. The choice of using angles (lat,lon) for metric data is confusing.*

In brief, the changes we make to the code now make it possible to run georeferenced, 10 bit geotiff datasets. When applying an orientation filter to the data (e.g. the NAOF we have implemented), we find little difference between 8 bit and 10 bit data even in shadowed/clouded areas. The orientation filter tends to cancel out contrasts within the data. The 10 bit data may prove advantageous where orientation filtering is not suitable and simple low-pass filtering and/or contrast limited histogram equalization (CLAHE) is used.

*The authors present a new image transform dubbed "NAOF", but is there a significant improvement, over for example typical*

*orientation filtering approach? At least to me, the 45 degrees filters seem like a redundant feature, as 90deg angled steerable filters incorporate all information [Freeman et al. 1991]. It seems the implementation is a combination of work similar to [Ahn et al. 2011], and orientation filtering [Heid et al. 2012]. But it is questionable, if the additional calculations add towards improvement. The filter banks are correlated (as just mentioned), furthermore, the visible bands are similiarly correlated over glaciers. Hence, the information gain might be very limited.*

We tested a number of different kernels for the orientation filter, and found that some of the simpler orientation filters would enhance and suppress features (particularly crevasse fields) depending on their orientation. The filter banks are correlated, but adding both the 90 degree and 45 degree filters preserves more feature uniqueness than either alone. This is important for reducing the number of false matches and signal to noise ration in crevasse fields, where subsequent crevasses have similar orientation signals. The visible bands are correlated, which is why we sum them into a single band. On some glaciers we have found that near infrared Band 8 of Sentinel 2 (842 nm) may produce a better feature contrast than other bands on some glaciers. We have also found that shortwave infrared bands 10 and 11 (1375 and 1610 nm) can be suitable in some cases (high contrast between ice and supraglacial debris), however suffer from a lower spatial resolution. Different band combinations can easily be created within SentinelHub (which we discuss briefly in the user manual). We mostly use contrast-enhancing SentinelHub custom script (javascript) which combines Sentinel 2 bands 3,4 and 8. We will add this custom script to a GitHub repository.

*The paragraph on velocity calculations talks about two methods to extract displacements from an image pair. However, this distinction is more about the option if refinement is applied or not. The authors limit themselves to frequency domain methods,*

*which is the de-facto procedure for PIV. However, spatial domain methods can be more favorable for glacier related applica-*

*tions. This comes down to the point given above, of the transfer from fluid mechanics towards an application domain without the ability to redo an experiment. Hence, the velocity profile can be highly variable (extensive shear, valley walls, diverging flow, fast flow, ...). Spatial domain methods are more resistant to shear flow, the chip is not related to the search space (as with fourier methods). The authors might have considered spatial domain methods, but for clarity it might be fruitfull to mention this, and why. In PIV experiments, one is able to adjust the image sampling rate, to be in accordance with the maximum flow*

*velocity, this is less the case for glaciers.*

At the early stages of this toolbox we investigated both frequency domain and spatial domain (Normalized Cross Correlation) matching algorithms. In particular we tested the NCC option within IMGRAFT (Messerli and Grinsted, 2015) and 'discrete cross correlation window deformation' option in PivLab (Thielicke and Stamhuis, 2014) which allows for shearing of features.

We found no clear improvement in matches relative to frequency domain cross correlation even in areas with significant ice shear (glacier margins with around 1000 m/yr velocity gradient across one kilometre). We also found that the multi-pass frequency domain matching would usually outperform single passes in both frequency and spatial domain. However, we found that changes to the pre and post-processing would often have the greatest improvement on final velocity maps, so have emphasized this aspect in GIV. This is indeed less of an issue within fluid dynamics where the sampling rate, tracers and illumination may be controlled.

We have added two sentences explaining why we opt for frequency domain methods in GIV, and some of the relative benefits of each method. We also refer readers interested in further discussion to Thielicke and Stamhuis (2014) and a chapter of a recent PhD thesis by Altena (2018) which provides an accessible and detailed discussion of these topics from the perspective of glaciology.

*Technical comments: In general the code is very messy, many duplicate lines exist, commenting is absent ("strcmpi(inputs{51,2}, 'Yes')"). At multiple locations indents are used in-appropriately. Documentation and commenting within the code is limited. Every function has an extensive help section, which is typically an acknowledgment and info about GIV. Input or output variables are not described. For many or all functions, tests are lacking. Factorization is on a low level. Given this situation of the*

*code, how do the authors see adoption by others? The objective of the authors is to be a wrapper (based upon ImGraft and matPIV), then one would expect documentation to be extensive.*

We would again like to thank the reviewer their time taken providing feedback on our code. We have inspected every function in GIV to improve formatting (tab only within loops, no unnecessary spaces, etc.), increase the number of in-code comments, and add to function descriptions. As mentioned above, reading the code is not necessary for running GIV (and is not possible in the case of the standalone apps). As such we do not believe code commenting is an important factor for the adoption of GIV by others. However, we aim for our code to be scrutable by those that do wish to read it, or wish to use our filters within their own code.

A number of the 'duplicate lines' within the code are present in order to enable the various input options. We hope that the additional comments throughout the code, alongside the change of our inputs array to a dot notation struct array will make these clearer. For example "strcmpi(inputs{30,2}, 'Yes')" should now read "strcmpi(inputs.parralelize, 'Yes')" and be accompanied with at least a brief comment.

*Sub-pixel: There does not seem to be a sub-pixel estimation present? Why is this?*

A sub-pixel estimation is present. For the single-pass algorithm it is built into the 'GIVtrack.m' function, while for the multi-pass algorithm a separate function ('GIVtrackmultipeak.m') is called within the final feature tracking pass ('GIVtrackmulti-final.m'). The multi-pass (recommended) sub-peak finder is set to a gaussian peak fit by default (and in the app), but can be modified to a centroid or parabolic fit within the code if desired. We have found that changing this has little effect on outputs.

*The domain of PIV is also extensively populated with papers about peak-locking, it would be good if the authors put some effort in this as well. This bias is especially present in sub-pixel estimation directly done on the correlation surface. Other methods, such as TPSS, as implemented in Cosi-Corr are less sensitive to this effect (which to a large extent might explain its popularity in geodetic imaging).*

We have not investigated peak-locking in detail while working on GIV. Inspection of raw displacement and velocity histograms shows only a minor bias towards integer values, and does not appear to be a major source of error.

A project is currently underway comparing feature tracking results from GIV and other feature tracking toolboxes to GPS derived velocity data. We are planning on considering the effect of different sub-pixel estimation schemes, and future versions of GIV will be updated if a particular algorithm is clearly superior. An in depth discussion of the particle tracking parameter options is beyond the scope of this manuscript.

*Geo-referencing: The gridspacing as it is implemented now assumes all rectangular and upright pixels, please adjust this.*

As mentioned above, we have added an option for GIV to process raw geotiff data and read in geographic data off the images.

If this comment is referring to the velocity fields being projected onto a horizontal plane (i.e. not corrected for topographic strike and dip), this is common practice in feature tracking. This correction may be applied using a DEM after running GIV, but is usually very small.

*Matlab: To my knowledge Matlab is not open access, the code might be open, but many of the algorithms are hidden away and licensing is needed. Hence, I am not so sure if the description about FFTW is needed, as one is not able to access this. Secondly, it is a standard routine within MATLAB.*

FFTW is an open source toolbox, however this is correct about much of MATLAB. The objective of this paragraph is to provide some justification for the toolbox being coded in high-level interpreted programming language MATLAB. FFTs constitute the largest computational expense of feature-tracking models, and in MATLAB are performed efficiently in C subroutine library FFTW rather than in any native MATLAB code. As such, GIV can process feature tracking pairs rapidly (particularly with the built-in parallelisation).

*Merging satellite data: The authors describe a time-series construction method, illustrate a toy example. But it seems they apply a sigma-filter, and do simple infilling. There are more advanced methods available, which are more adaptive towards velocity data. Hence, a simple reference to (e.g.: [Mouginot et al. 2017]) might suffice for the implementation here, and this section can thus be reduced considerably.*

*The temporal sampling is a bit counter intuitive, as this is an opposite direction to the workflow of [Millan et al. 2019]. Where they found the time-span needs to be of sufficient size, in order to be of most use. Why then do the authors prefer the shorter time steps?*

We do not fully understand the above comments. Our findings on optimal temporal sampling are in line with Millan et al., 2019, and we by default exclude velocity image pairs with temporal separation of less than one week. We describe the example of a very slow moving glacier in the Chimborazo case study, where excluding time steps shorter than 6 months provided the best velocity results. The minimum and maximum time separation limits can be adjusted within the user interface according to the specific characteristics of the glacier of interest.

If this comment is referring to the first iteration of monthly timeseries generation, the 0-1 scoring system refers to the proportion of a velocity map within a given month rather than the temporal separation. A 7 day or 30 day separation velocity map entirely within a given month will both be assigned a score of 1. Velocity maps overlapping into other months will be assigned lower scores.

The sigma-filter and infilling are post-processing steps to improve the quality of individual monthly maps.

Other averaging and timeseries generation methods for glacier velocities are available (e.g. Millan et al., 2019's NetCDF Geo-Cubes, Altena et al., 2019's Hough space method, etc.), although do not inherently generate monthly time-series. Our method generates a full dataset mean and median, as well as monthly velocity maps which (we hope) are easier to interpret than the unevenly sampled satellite image pair timing.

*Code specific: There is a function about intensity capping, while this might be of interest to PIV, it is questionable if this is of interest to environmental signals. Bright intensity of seed points within dark water are very much different from glacier surfaces. Also how does this merge with the high-pass filtering... ?*

Intensity capping is not usually necessary, but can be useful where bright snowpatches are present close to a debris cov- ered or otherwise dark glacier. In general the filters do not need to all be applied or stacked. By default only the orientation filter is applied. My recommended filter choices are:

- Orientation filter (NAOF) only = default

- High-pass filter and CLAHE (+ Sobel filter in some cases) = where orientation filtering creates too much noise

- Intensity cap + High-pass filter and CLAHE (+ Sobel filter in some cases) = where bright patches are present, and orientation filtering creates too much noise

Users may wish to experiment with a small dataset, different glaciers will have different optimal pre-processing parameters.

*Even though your algorithms are optimized, they might not pull out the best of the machinery, yet. For example, the function "neighbourfilter" contains a double loop, to process a kernel. An internal function of Matlab called "nlfilter" might*

*be much faster. A good reference for such implementations can be found in "Accelerating MATLAB Performance" (http: //undocumentedmatlab.com/books/matlab-performance). In the same function from line 95 onward, the authors use double indexing. While linear indexing is possible as well, which make vectorization possible (which is MATLABs strong point). This greatly improves the processing time as well.*

We thank the reviewer for the useful reference and recommendations. We test an implementation of 'neighbourfilter.m' using nlfilter, however it results in little overall speed-up and some unexpected artefacts. The artefacts can probably be fixed by re-writing the function around nlfilter, however due to the small overall increase in efficiency we leave the function as is for now. We believe that the inclusion of parallel computing of feature-tracking pairs is likely to result in the greatest computational speed-up relative to other tools.

*I am not sure if the variance calculation of the flow direction is calculated correctly. Offsetting the direction and applying a weighting might work, but maybe correct circular statistics might be more appropriate, see [Berens 2009] for a toolbox implementation. It might be more easy to use the mapping coordinates instead...?*

Our flow direction mean calculation appears give the same results as Berens's CircStat on examples tested, but are not sure of its behaviour in all scenarios. Thus, we replace it with the well-tested CircStat.

We make several changes to CircStat to enable calculation of mean and standard deviation in data containing Not a Number (NaN) values. We will share the NaN tolerating version of the toolbox on GitHub or file exchange.

*It might be good to give an example; here is a sniplet of your code (a straight copy):*
*if smoothsize == 2;*

*mask = [0 1 0; 1 0 1; 0 1 0 ];*

*elseif smoothsize == 3;*

*mask = [0 1 0;1 1 1; 1 0 1; 1 1 1; 0 1 0 ];*

*elseif smoothsize == 4;*

*mask = [0 0 1 0 0 ;0 1 1 1 0 ;0 1 1 1 0; 1 1 1 1 1 ; 0 1 1 1 0; 0 1 1 1 0; 0 0 1 0 0 ];*

*elseif smoothsize == 5;*

*mask = [0 0 1 0 0 ;0 1 1 1 0 ;0 1 1 1 0;1 1 1 1 1 ;1 1 1 1 1 ; 1 1 1 1 1 ; 0 1 1 1 0; 0 1 1 1 0; 0 0 1 0 0 ];*

*end*

*why isn't this in a function? Something like: function [mask] = make_mask(smoothsize)*

*"*

*generates a neighborhood kernel in the form of a diamond,*

*though excludes the central pixel*

*input:*

*smoothsize - integer value*

*output:*

*mask - logical array*

*"*

*if nargin<1, smoothsize = 3; end*

*mask_radius = floor(smoothsize/2);*

*mask = strel('diamond', mask_radius); % make a diamond shape*

*mask = mask.Neighborhood;*

*mask(mask_radius+1, mask_radius+1) = 0; % exclude the central element*

*end*

That function is indeed much more elegant and flexible than our implementation. We have implemented it in the code (with due acknowledgement) and hopefully it makes the filter more flexible for other uses.

*In all I am aware this is, like all research, work in progress, and I think this is a very useful direction. But in its current state and form, sufficient work needs to be done to be of interest, please see [Perks 2020] for a nice example. Here workflows are nicely presented, a dashboard is present, etc. All in all, I think a transfer towards a technical journal of EGU might be more in place.*

We hope that the comments and code edits presented clarify our decision to submit to The Cryosphere, and answer some of the concerns presented in this review.

*Textual: p63: "broken down" is misleading, as overlapping chips can also be used p140: superscript the 2, to make it a square*

The recommended changes have been made.

---

## Author Response (AR3)

**Glacier Image Velocimetry: an open-source toolbox for easy and rapid calculation of high-resolution glacier-velocity fields**

Maximillian Van Wyk de Vries1,2 and Andrew D. Wickert1,2

1Department of Earth & Environmental Sciences, University of Minnesota, Minneapolis, MN 2Saint Anthony Falls Laboratory, University of Minnesota, Minneapolis, MN

**Correspondence:** Maximillian Van Wyk de Vries (vanwy048@umn.edu)

**Author Response**

Contents of this pdf:

- Response to editor
- Response to reviewer
- Supplementary experiments 1 and 2

**Response to editor:**

Comments to the Author:

Dear authors,

Your manuscript has been reviewed a second time by one of the original reviewers. This reviewer remains very critical about your work, and still sees major issues that prevent her/him to recommend your manuscript for publication (in fact, the anonymous reviewer suggests a "rejection" at this point). Given the contrasting views on the manuscript, with one reviewer (Ted Scambos) supporting your submission and the other one (anonymous) being very critical, the review process will be continued.

I first invite you to answer all new queries raised by the anonymous reviewer, after which the manuscript will be sent out for review again. In answering the issues raised by the critical reviewer, it is important to go in detail and provide quantitative information where possible. In particular, pay attention to the comment by the reviewer stating that: "Secondly, the response by the authors is brief and no effort is put in backing-up their argument. The majority of answers include phrases like, "we find little difference", "we found no clear improvement", but results or information is lacking. Thus it is difficult to get convinced, especially since the points I have raised are backed up by references. One can not expect me to run the test for you, in my opinion the burden of proof needs to be at the side of the authors". When answering this question, I would also like you to expand on some of the answers you previously gave, by providing additional quantitative information (and update the manuscript accordingly) on:

• "we find little difference between 8 bit and 10 bit 70 data even in shadowed/clouded areas. The orientation filter tends to cancel out contrasts within the data. The 10 bit data may prove advantageous where orientation filtering is not suitable and simple low-pass filtering and/or contrast limited histogram equalization (CLAHE) is used

o What is 'little difference'?

• "On some glaciers we have found that near infrared Band 8 of Sentinel 2 (842 nm) may produce a better feature contrast than other bands on some glaciers. We have also found that shortwave infrared bands 10 and 11 (1375 and 1610 nm) can be suitable in some cases (high contrast between ice and supraglacial debris), however suffer from a lower spatial resolution"

o 'may produce better feature contrast': by how much? And can you specific about the resolution where the problems originate for the shortwave IR?

• "We found no clear improvement in matches relative to frequency domain cross correlation even in areas with significant ice shear (glacier margins with around 1000 m/yr velocity gradient across one kilometre). We also found that the multi-pass frequency domain matching would usually outperform single passes in both frequency and spatial domain. However, we found that changes to the pre and post-processing would often have the greatest improvement on final velocity maps, so have emphasized this aspect in GIV. This is indeed less of an issue within fluid dynamics where the sampling rate, tracers and illumination 110 may be controlled."

o You discuss the (lack of) 'improvement', several times. Can you be more specific / quantify the (lack of) improvement

• "We test an implementation of 'neighbourfilter.m' using nlfilter, however it results in little overall speed-up and some unexpected artefacts. "

o Can you quantify the speed up and describe the unexpected artefacts

The comment about the suitability of the journal ('The Cryosphere') has been addressed sufficiently in my opinion, and will not be a point of debate in the final decision (as I already indicated before). I agree that the study could have been considered for a more technical journal, like Geoscientific Model Development, but I am still convinced that The Cryosphere is a suitable journal (given that the study fulfils the scientific standards of this journal, which needs to be further assessed through peer review).

Thank you for your efforts,

Best regards,

Harry

Dear Dr. Harry Zekollari,

We thank you for the comments above, and for the opportunity to respond to anonymous reviewer 2's comments. We have designed two additional experiments to quantitatively back up our answers to the reviewer's questions, which are presented in the response below and will be uploaded to the supplementary materials of our paper. These experiments are designed to evaluate the relative quality of velocity maps derived from different processing parameters and Sentinel 2 image bands.

We update the manuscript to include mentions of this additional information, although it is largely outside of the main objective of our manuscript. We are not aiming to provide an evaluation of different parameter options and choices in feature tracking, but rather to present a new feature-tracking toolbox to the glaciology community. We worry that making major modifications to the manuscript towards the latter of these objectives (as the reviewer appears to suggest) may detract from the first objective. In order to fully evaluate the accuracy of feature tracking under different parameter choices, comparison of model results to ground-based datasets (e.g. GPS and/or ground based camera datasets) would be ideal. An effort is currently underway to compare different toolboxes (GIV, vmap, CARST, Auto-RIFT, possibly more) to a glacier-surface GPS dataset, which will be build on the work conducted in this manuscript.

We hope that our detailed response helps convince reviewer 2 of the benefit of the filters and methods presented in this paper and available in GIV. The reviewer's most severe criticism relates to the choice of TC as a venue for this paper, which we hope is mitigated by your comments on the matter. We strongly disagree with the characterisation of GIV as a "straightforward copy" of Thielicke (2014)'s toolbox. While some of the core computational techniques are similar, neither the objectives nor code of GIV are alike. Minor sections of the PIVlab toolbox are included (and clearly acknowledged) in GIV, but these make up

---

## Author Response (AR4)

**Glacier Image Velocimetry: an open-source toolbox for easy and rapid calculation of high-resolution glacier-velocity fields**

Maximillian Van Wyk de Vries[1,2] and Andrew D. Wickert[1,2]

[1]Department of Earth & Environmental Sciences, University of Minnesota, Minneapolis, MN
[2]Saint Anthony Falls Laboratory, University of Minnesota, Minneapolis, MN

**Correspondence:** Maximillian Van Wyk de Vries (vanwy048@umn.edu)

**Author Response**

Contents of this pdf:

- Response to editor
- Response to reviewer 3

**Response to editor:**

Comments to the Author:

Dear authors,

Your manuscript was now reviewed by a new/external reviewer. This reviewer has several (some rather technical) comments, which I now invite you to address in detail (point-by-point rebuttal + manuscript update in accordance). This includes a comment related to the suitability of TC as a journal, which I also ask you to answer for the record. As previously indicated, as an editor, I understand a part of the critique by the reviewers, but I nevertheless also think that TC is a suitable option and therefore support your choice.

Many thanks for your efforts,

Kind regards,

Harry

Dear Dr. Harry Zekollari,

We thank you for the comments above and have responded to the reviewer's comments in detail below. In particular:

- We have responded to the reviewer's comments about the choice of TC as a journal, laying out in more detail our objectives with this manuscript and why we think they are of interest to TC's readership.
- We have run a new test on a synthetic image to highlight some of the characteristics of the new image pre-filter (NAOF) and explained our reasoning while creating this filter.
- We have repeated the experiments in the supplementary materials with a different method (orientation correlation) in response to a query about this.

The manuscript has also been updated accordingly, as shown in the tracked changes version. We also update the acknowledgements to include the third reviewer – comments by yourself and all three reviewers have helped improve the manuscript and code, and refine the objectives of this paper.

M. Van Wyk de Vries and Dr. A. D. Wickert

**Response to reviewer (3):**

We thank the reviewer for the detailed comments and suggestions, which we have answered between the lines below.

Overall, I think the this is a useful tool (i have not actually tested it). However, there is not much which is new beyond the tool itself, and it seems like it is misplaced in TC. I feel like this is an editorial decision though. Atleast it targets the correct audience.

In essence, we have three primary objectives with this paper:

1. Provide background on past advances and the current state of feature tracking in glaciology, and why these methods are valuable.

2. Present a new, open-source feature tracking toolbox, GIV, and explain in accessible terms the advances of this toolbox (handling large timeseries, new image pre-processing and velocity map post-processing, etc.), and its objectives (speed, flexibility, and ease of use on laptops and personal computers).

3. Examine the outputs of this toolbox through a variety of glacier types, both to confirm its pertinence in a range of scenarios and evaluate the type of glaciological problems which it may help solve.

Based on these objectives, we believe that this paper will both be relevant to the scope of The Cryosphere, and of interest to its readership. We agree that some of the computational methods described are not new- feature tracking has been in use in the earth sciences for several decades.

However, GIV does combine new methods of pre and post processing imagery, as well as convenient methods for handling large timeseries – with no coding required - in glaciology.

To provide one example where we think GIV can be particularly useful, it can be used to rapidly process a long timeseries of velocities following a localized natural disaster (e.g. landslide, glacier detachment/surge, etc.). Users with no prior experience with feature tracking may obtain a monthly velocity timeseries (using several hundred image pairs) in only a few hours of work, for a rapid evaluation of the hazard and prior conditions.

In equation 1 you write a sum over α. Can you please write it as a sum over i=1 to 4 and then write α_i inside the sum.

We have updated the equation and associated code to instead read:

$$I_f = \sum_{i=1}^{4} \cos\left[\arctan 2(I_o * \alpha_i, I_o * R[\alpha_i])\right]$$

I have some issues with the NAOF filter in equation 1. I appreciate the idea of extracting angles using multiple filtering kernels in order to get a more robust estimate of the local gradient. However, I miss an explanation or theoretical justification of why you believe this filter is good. Why is it constructed the way it is? I need some motivation for it.

The name of the NAOF suggests that this is similar to the filter used in orientation correlation (OC). But in the OC filter the output is a complex number which preserved the local gradient direction. Here you take the real part and any information about the gradient orientation is lost.

In equation 1 you convolute with four different α's. Each rotated by 45 degrees. But since each arctan2 operation uses both α, and R[α], 2 out of those 4 convolutions seem to me to be giving no additional information. Here's how I imagine the orientations of the different filtering kernels to be (based on your description). In order to help visualize it i define a matrix: a·α+b·R[α] - That gives me these four matrices.

matrix1 = [+a 0 -b;0 0 0;+b 0 -a]

matrix2 = [0 -b 0;+a 0 -a;0 +b 0]

matrix3 = [-b 0 -a;0 0 0;+a 0 +b]

matrix4 = [0 -a 0;-b 0 +b;0 +a 0]

So, you calculate the two gradients in the a and b-directions. Then you plug that into exp(i*arctan2(..)). That gives you a complex unit vector in that direction. Then you throw away

the imaginary part and project it down on the x-axis. -But consider how the imaginary part of the filtering associated with matrix1 is almost identical to the real part of the matrix3 situation. This wastes multiple convolutions and slows the prefiltering down. Further, I do not understand the motivation for taking summing the x-component of 4 different unit vectors representing the intensity gradients in different coordinate systems. This seems very arbitrary. There is no explanation for why the NAOF prefilter is constructed in this way. This suggests to me that there is no theoretical justification or motivation, and that this new NAOF prefilter is an ad-hoc construction.

Note also: that real(exp(i*atan2(6,4))) is the same as cos(atan2(6,4))

Without a clear and convincing motivation for the form of the equation I find it difficult to understand why it is named "near anisotropic orientation filter". The filter is demontrated to be pretty good in practice in the selected examples. That is nice. -But is it an advancement over existing methods (CLAHE+H, or Orientation Correlation prefiltering)? I am not convinced that it is better in general.

Many thanks for these comments. We have made a number of changes to the text, and moved some details about the filter to the supplementary materials to not emphasize it more than required. We did not mean to present the NAOF as the 'perfect' pre-filter for use in feature tracking. We simply are presenting it as a viable alternative, which in some cases produces less noisy velocity maps than the other filters included in GIV. Its effectiveness has mostly been tested empirically. We also include other common filters such as CLAHE, highpass, Laplacian and Gaussian, such that users may test image filter combinations depending on the local glacier characteristics.

The objective for the NAOF was to:

  1) Be insensitive to feature orientation.
  2) Retain feature uniqueness.
  3) Not shift the location of features within the image(s).
  4) Remove contrasts between areas of differing pixel intensity (cloud cover, shadows, etc).
  5) Retain the image as a real number matrix with only one band (two dimensional).

We hope that the improvements to the wording of this paragraph clarify our design and use of this filter- which was calibrated based on empirical tests. As the reviewer notes, the NAOF is based on the filter used in orientation correlation (OC). Rather than retaining the real and imaginary components, we retain only the real component and sum across four different filter angles.

The reasoning for editing the orientation filter used in OC is that despite producing good results, it was sensitive to feature orientation for many filter kernel choices. For a filter $\alpha = [1\ 0\ -1]$ and the formula for calculating an orientation image given below:

$$I_{OC} = \exp\left(i \times \arctan 2(I_o * \alpha, I_o * R[\alpha])\right)$$

both the real and imaginary portions of the resulting image are more sensitive to certain feature orientations than others. On glaciers the features tend to be highly directional (crevasse fields), and this can result in a degradation of feature tracking results for portions of the glacier with surface features oriented in-line with the filter. In addition, certain undesirable features (e.g. medial moraine) can be emphasized over local crevasse fields, reducing the number of correct matches. Therefore, we decide to combine the four orientations, rather than using a single oriented filter. Certain filter kernels (e.g. [-1 1]) are not centered on the feature, and so do not preserve feature location.

Running some tests on example images showed that combining the four directions into one single filter (e.g. filter kernel [-1 -1 -1; -1 8 -1; -1 -1 -1]) provided adequate edge/feature enhancement, however resulted in a loss of feature uniqueness and a much greater number of false matches. Calculating the four components separately however allows each to be weighted differently according to the original pixel values, and better preserve feature uniqueness. Figure RR1 below shows the filter results for a synthetic image example, and figure RR2 shows the resulting filtered values plotted on a histogram. NAOF both detects all line orientations, and provides a broader distribution of resulting pixel values than the other single kernels.

Based on the comments here, we have rewritten NAOF to be more computationally efficient. We now pre-calculate each filtered image and 2-argument arctangent. Commuted 2-argument arctangent values are calculated as arctan2(b,a) = pi/4 - arctan2(a,b), reducing the number of operations. None of the convolutions are redundant, as each is sensitive to a different feature orientation. See figure RR3 in which a portion of the 2020/01/01 Sentinel-2 image of Glaciar Perito Moreno is shown under the different filter constituents and full NAOF.

[Figure]

Figure RR1: Resulting filtered images for a small portion of the synthetic image above. Note how the [-1 0 1] kernel loses information on some horizontal features.

[Figure]

Figure RR2: Histograms of filtered pixel values for the three filtering options and original image shown in figure RR1. Note how NAOF produces the largest spread of unique values, whereas the [-1 -1 -1; -1 8 -1; -1 -1 -1] produces only a very limited range of values.

[Figure]

Figure RR3: 2020/01/01 Sentinel-2 image of Glaciar Perito Moreno shown under the different filter constituents and full NAOF, and full NAOF. Note how orientations 1 and 4 are particularly sensitive to the longitudinal features (medial moraine + flow bands), while orientation 3 clearly delineates the local crevasse field. Note also how crevasse uniqueness is preserved in the final summed filter.
* * *
Line116: It is fine to include a reasonably good new prefilter approach even if it is not theoretically justified. But I dont think it is a major step forward without more theory behind it, and/or a much larger test dataset. So, I recommend that you are conservative in your claims as to how good this new prefilter is. E.g. "NOAF shows comparable performance to CLAHE+H (see supplement)".

We have adjusted the description of the image filters in the text, and do not present NAOF as superior to other filters in all circumstances. Our evidence for the value of NAOF is mostly empirical. We do nevertheless note that it has produced the best velocity maps over a range of glaciers.

From the supplement the new NAOF performs similar to CLAHE, except CLAHE seems to almost fail when bitdepth=12. That to me seems weird. (Is it because CLAHE is extremely sensitive to noise in the least significant bits ). Please explain why CLAHE+H is almost broken when bitdepth=12 if you can. If this is a real problem with CLAHE, then I think you should highlight it.

We have repeated the experiment as it appears that CLAHE was not correctly applied to the 12-bit image pairs. The supplementary materials have been updated to reflect this (12-bit and 8-bit CLAHE produce very similar results).

In the supplement you show some examples of NAOF, CLAHE, Raw, and combined with FCC and NCC and different bit depths. The issues with NCC+Raw are well known. ImGRAFT has Orientation Correlation as the default option, which presumably would work much better. This makes me ask why have you deliberately chosen to compare to NCC instead of Orientation Correlation.

We used NCC in response to some particular questions by the previous reviewer (anonymous reviewer 2). For completeness, we have also repeated the same experiments with CCF-O (Orientation correlation) option of IMGRAFT and the same parameter options as used for NCC. This produces improved results for the raw imagery (as expected based on prior findings), and similar results to NCC for CLAHE+HiPass and NAOF filtered results. Running NAOF filtered images with orientation correlation may 'over-filter' the image, as it effectively applies a double orientation filter. See figure RR4 below.

[Figure]

Figure RR4: Results of experiment 1 (see supplementary materials) updated to include orientation correlation results.

Line 108-110: You say this is better than a applying a single filtering kernel. This statement should be weakened if you do not provide data to demonstrates this. (That would be difficult because there are so many ways you could design the filtering kernel.) So you probably have to content with saying it is better than some specified filtering kernel.

We have adjusted the wording of this sentence to highlight which single filtering kernel we are referring to. Our objective was to state that the multi-orientation NAOF approach produces better results than the single filter summing the multiple oriented filters together (e.g. see figure RR1 and RR2).

Section 2.2: This section distinguishes FFT vs spatial domain methods. But in my view the FFT methods are really just a faster implentation of the convolutions.

We have made some adjustments to this section to clarify the key points. Spatial domain convolution and Fast Fourier Transform convolution can be equivalent (as per the convolution theorem), however they are used in different ways. Spatial convolution is used for matching one small region to surrounding small regions using a sliding-window approach, while frequency correlation is used to compute the match between two regions in one single calculation. As such, they do ultimately result in differing correlation surfaces- see Figure 4.9 from Altena (2018) for one example (NCC vs FCC correlation surface). Thielicke and Stamhuis (2015) also have a

description of this in page 3 of their paper. We hope that there is less confusion in this section after the adjustments.

Line 347: "incorporates" is not a good word here, maybe "is able to exploit" would be better. I dont think you can claim credit for the new data availability.

We agree and have adjusted the wording to 'builds upon'. We do not claim credit for the availability of satellite imagery.

Line 162-174: It is much safer to do statistics and outlier rejection and interpolation steps based on the x and y component velocities rather than on |V| and flow direction. -line 172 makes me uncomfortable. Please clarify. Here's a little more detail of what i mean: "mean(sqrt(vx^2+vy^2))" will tend to be greater than "sqrt(mean(vx)^2+mean(vy)^2)" if there is any noise in vx and vy. The latter is more correct.

This is a very good point, and is key to obtaining good velocity maps in some cases (particularly with extremely slow moving glaciers). We do outlier rejection based on both x-y component data and speed/flow direction.

For instance:

-The stable ground correction is applied in the x and y components

-Average velocity maps are calculated based on averaging of x-y components "sqrt(mean(vx)^2+mean(vy)^2)" & "sqrt(median(vx)^2+median(vy)^2)" (some small adjustments made to the code to ensure that monthly values, etc are also calculated this way)

-A maximum speed filter is applied (from "sqrt(vx^2+vy^2)").

-Flow direction filters may also be applied.

Supplement

The SNR definition as written is dangerous to apply in practice. E.g. C_NCC can be both positive and negative values. therefore in some rare cases mean(C) can end up being negative. mean(C) also includes the peak which will bias the Sis not NR ratio as the "noise" now contains some signal.

Indeed, we had missed out the absolute value symbol in the equation. SNR is calculated as [peak/abs(mean(C))]. Note that the 3x3 pixel area surrounding the peak (used in subpixel peak finding) is cropped out of the correlation matrix prior to calculating the mean.

For that reason i think it is better to estimate the noise level using something like median(abs(C)). abs to deal with negative. Median is more robust to outliers (=peaks).

As mentioned above, the primary peak is cropped out of the correlation surface. Both mean(abs(C)) and median(abs(C)) have their advantages, as sensitivity to outliers can be desirable. Due to the variety of formulations, care must be taken when comparing SNR scores, particularly between different codes.

Figure S5: For the different methods (FCC/NCC) the C entering the SNR calculation has a different definition. Therefore you cannot make a straight comparison between SNR_NCC and SNR_FCC. E.g. Imagine i now define a new matching metric I called KFC. In my KFC method I simply calculate C_KFC = exp(C_NCC). It will have exactly the same peaks just mapped through a nonlinear transformation. However, the SNR will be completely different.

This is true; however KFC is not really a different matching metric and this example presents a worst case scenario. The signal to noise ration and peak ratio in FCC and NCC do result from different calculations, but these are not nonlinear transformations. In each case the signal to noise ration and peak ratio represent the same property ('how confident are we that this peak is a real match, rather than an artifact'). Using ratios (as opposed to raw correlation scores) and both the peak and signal to noise ratio reduces the dependence on the matching method.

Other comparison metrics exist, such as the 'percentage of correct matches' used by Heid and Kaab (2012). This is very sensitive to the methods used to evaluate correct matches, particularly where no external velocity results are available. Calculation of velocities over stable ground can also be used, but is potentially sensitive to the matching method and does not always represent the quality of matches over the glacier.

In an ideal situation, we would evaluate the 'correctness' of each matching technique with reference to external, ground-based glacier velocity measurements. This comparison should ideally include other feature-tracking codes (e.g. AutoRIFT, CARST, IMGRAFT, etc.), and is beyond the scope of this paper.

In summary, SNR and PKR are great metrics for comparing different options (e.g. image filters) across one feature tracking method. SNR and PKR are not perfect for comparing between feature tracking methods, but (particularly in combination with each other) do provide useful information about the relative quality of matches. We apply them with caution here, and have added a brief note to the limitations paragraph.
* * *
Figures: The number of figure are a bit excessive to me. E.g. fig 10 looks cool, but what does it bring to the manuscript. Fig11 seems unnecessary too.

Figure 10 shows the velocity results over a small, slow moving glacier system (which is typically challenging to resolve via feature tracking). Figure 11 shows GIV's graphical user interface, through which all calculations can be made. We do think these figures are useful, complement the manuscript text, and will be particularly relevant to The Cryosphere's glaciological audience.

References:

Heid & Kaab, 2012. 10.1016/j.rse.2011.11.024

Thielicke & Stamhuis 2014. 10.5334/jors.bl

Altena 2018. https://www.duo.uio.no/handle/10852/61747

---

## Author Response (AR5)

**Glacier Image Velocimetry: an open-source toolbox for easy and rapid calculation of high-resolution glacier-velocity fields**

Maximillian Van Wyk de Vries[1,2] and Andrew D. Wickert[1,2]

[1]Department of Earth & Environmental Sciences, University of Minnesota, Minneapolis, MN
[2]Saint Anthony Falls Laboratory, University of Minnesota, Minneapolis, MN

Correspondence: Maximillian Van Wyk de Vries (vanwy048@umn.edu)

**Author Response**

Dear Dr. Harry Zekollari,

Many thanks for the detailed comments. You have raised some important points about the text and figures, which we have carefully addressed in the final version of the manuscript submitted alongside this response. We have responded to each comment between the lines below.

Dear Maximillian Van Wyk de Vries and Andrew Wickert,

Many thanks for sending in a new version of your manuscript and accompanying rebuttal letter. Your manuscript is now in a good shape and almost ready acceptance. I have formulated a list of mostly minor and technical comments that I would like you to address when resubmitting your manuscript, before proceeding to a final acceptance. Pay particular attention to the remarks related to the figures, which will need some reworking in some cases:

- l. 4: "...and often a safety risk": sounds a bit odd to me. Maybe change to: "...and often associated with safety risks"

We have edited the text based on this suggestion.

- l. 6: "...hence, velocity over time.": just to make this entirely clear, suggest referring to "...hence, surface velocity over time"

We have added this clarification to the text.

- l.8-10: please be consistent when referring to glaciers and ice caps. You should refer to all ice bodies as glaciers (in line with IPCC) or instead consistently refer to certain ice bodies as ice

caps (in line with for instance EGU, which mentions 'ice caps' in session titles and has a 'science officer for ice caps'): you now refer to Vavilov ice cap as a "very large glacier". Probably easiest to not refer to ice caps (in line with IPCC): i.e. "...as well as a tropical glacier (Volcán Chimborazo)". Please make sure that this is consistent throughout the manuscript.

Many thanks for raising this potentially confusing point about terminology. We have removed all mentions of 'ice caps' for both Vavilov and Chimborazo and replaced them with the term 'glacier'. We have retained the name 'Vavilov Ice Cap' as it is used in prior literature, and ensured it is capitalized throughout.

- introduction: is really very broad and elaborately touches upon topics that are not treated at all in this manuscript. You can have a general introduction, but it should become more compact: l.15-37 should ideally be compacted to max 10 lines, where the only part really relevant here seems to be around l. 28-32 + l. 35-37 (add references for these statements)

We have re-arranged the first three paragraphs, removing the less relevant material and consolidating them into one single paragraph ~12 lines long. The paragraph could be further shortened by reducing the number of references, but we hope these will provide useful context for any reader seeking additional details on this background.

- l. 31-32: 'Wal et al., 2008' should be 'van de Wal et al., 2008'. Could also refer to some more recent works that focus on remotely sensed velocities on glaciers here, as this is the main topic of your paper (vs. ice-sheets): e.g. Altena and Kääb (2017, Frontiers in Earth Science), Altena et al. (2019, The Cryosphere) and Dehecq et al. (2019, Nature Geoscience).

We have added Millan et al 2019 and Altena et al. 2019 to provide some recent context, and corrected van de Wal et al., 2008's citation. I should have noticed this given that my own name often ends up as 'Vries M.V.W.D.' in these lists…

- l. 76: "...and lighting conditions depend strongly...": maybe also explicitly mention the shadow(ing)?

We have added a mention of shadowing to this line.

- Table 1: very useful table for the reader! A few remarks. For EMT: any indication about the environment/language in which the 'worklfow' is? Should 'matpiv' not be 'MatPIV'? I found it a bit strange to have the explanation in the caption go from 3 to 1: maybe change order from 1 to 3?

We are glad that this table is helpful. EMT is distributed as precompiled binaries (e.g. https://wwwpub.zih.tu-dresden.de/~photo/emt/index.php) and I could not find a mention in either the paper or website as to the language these were originally written in (C++ is my best guess). We have updated the numbering to run from 1 to 3 rather than from 3 to 1.

- Figure 1: width of the individual boxes seems a bit random: why having them smaller in steps 1-3-4, then wider for step 5, and then smaller again: suggest having the same width (except step 5 maybe, as you want to emphasize this I guess? Maybe here also explain in caption why this is

shown in bold) or having different widths to have content on single line (e.g. for step 8 and step 9)

We have adjusted the formatting in this figure such that the different boxes are now the same width, except for the feature tracking step which we leave in bold for emphasis. This is no explained in the figure caption.

- section 2.2: was easy to follow in general, also as a non-specialist. I was wondering if the readability would not further improve by subdividing this section in several subsections, as quite different things are treated here: e.g. l.118-131: frequency-domain matching, l. 132-144: single- and multiple pass approaches, l.145-148: move this to part where FFT is introduced, l.149-156: parallel computing, l. 157-161: non-consecutive imaging, l. 162-178: accuracy assessment and final velocity map improvement, l.179-200: temporal resampling through iteration, l.201: georeferencing

We like the suggesting of adding subsections to this part of the paper, and have added them in (and rearranged the text accordingly).

- l. 139: 'matpiv' (two occurrences) should be 'MatPIV'

Thanks for catching this, we have updated matpiv to MatPIV throughout.

- Figure 2: nice visualization! Could you indicate which area/glacier we are looking at here? Add this information in caption, and potentially complement with coordinates (lon-lat) in image.

The images are taken from Amalia Glacier in the Pacific side of the Southern Patagonian Icefield. I have added the name and lat-lon of the images to the figure caption.

- l. 149-156: very nice to read about this option to parallelize the code. Small question here, which may be related to my misunderstanding: how are you accounting for displacement that occurs between the different image pairs (which are treated on different cores)? Maybe add a short 1-2 sentence description of this, or explain how this is not a problem (I guess others may have the same question?)

I am not sure if I fully understand your question here. The image pairs (both consecutive and non-consecutive) are assembled prior to calculation of displacements between them. The displacements are them calculated on different cores, and re-assembled into one single output matrix (storing all velocity outputs) following calculation. This way the large output matrix is not passed to individual cores (causing memory issues and slowing performance), but the rate limiting step of calculating displacements can be performed on multiple image pairs simultaneously. The output matrix of all velocities (MATLAB array named 'images', for reference for anyone looking at the code) is then post-processed as described in the rest of this section.

- l. 177: "if the dataset is smooth enough to allow it": how is this determined? Criterion for this? If so, could you give a short indication about how this is done?

We have removed this part of the sentence, as it is confusing and not necessary. We were referring to the fact that data cannot be interpolated if a too large number of 'not a number' (empty) values are present. The 'smoothness' of the surface is not calculated at this stage (although filters based on local standard deviation are used, as described in the text).

- Figure 3: please do replace the red or green color by another: is problematic to have both colors for line data for color-blind people (deuteranopia) (I noticed that you took this into account for other figures as e.g. mentioned in l.203-204: nice!). In panel b: how is the speed-up defined? As a fraction I guess? If so, probably more intuitive to formulate as % speed up.

We have replaced the color scheme with a colour-blind friendly qualitative color scheme from ColorBrewer (https://colorbrewer2.org/#type=qualitative&scheme=Dark2&n=3). We have also changed speed-up to % speed up for clarity.

- l.190-192: found this sentence difficult to follow as it is lengthy + use of numbers for weight, while 'one' is used to refer to velocity as well. Consider splitting sentence in two and reformulating second part for clarity.

We have re-written this sentence for clarity, it now reads "The weighting parameter is determined by the proportion of the individual map contained within a given month. For instance, a velocity entirely within one month will be weighted 1, while a velocity spread evenly over four months will be weighted 0.25."

- l.195-196: "Outlier detection and maximum velocity filters are implemented": could you provide some (even compact) information about how this is done / based on which criteria?

We have split this sentence into two and provided additional details. The maximum velocity threshold is the same as that used for the initial velocity calculations, and is defined by the user in the GUI.

- l.197-198: " ... are not be adapted..." should be "...are not adapted..."

Thanks for catching the typo, we have corrected this sentence.

- Figure 5a: change the color scheme to one that is suited to represent sequential data! The color scheme that you use here is suited for diverging data. See e.g. https://colorbrewer2.org/ This is of large importance, as you refer to this as: 'generation of publication-quality images of the velocity and flow direction maps'. I agree that the figure looks nice, but the color scheme you use does not align with your purpose.

This is an important point, thanks for picking it up. We have replaced the diverging colour scheme with a suitable sequential color scheme for the figure, and made the same change within the code. We have also edited the relevant function within the source code (save_images.m) such that users can easily select other preferred colormaps with only a single change in the code. We have set the default to be Crameri's 'Batlow' sequential color map, which should be appropriate for most conditions. Crameri's 'lajolla' and 'oslo' color maps also work well, as well as ColorBrewer's 'YlGnBu' sequential map. (The cyclical color map for the flow direction plots can now also be changed at the same point in the code).

- l. 219: write out the two occurrences of '3': 'three'

- l. 221: 'based off of...' : should this be 'based on...'?

We have made both changes to the text.

- figure 6: in the text you refer to the real names of the glacier (Spanish), while in the figures you use the English names: be consistent. For the location map: would it maybe make sense to have this a bit more focused on Patagonia? I guess most people know where Patagonia is, but have little idea where the two glaciers are in Patagonia (e.g. are they 20 km apart from one another or 500 km?)

We have modified the glacier names to the original Spanish, and added a small inlay of the Southern Patagonian Icefield to show the location of the two glaciers discussed.

- figure 7: same remark as for figure 6 concerning the location map. Where in the European Alps is this glacier vs. where are the European Alps.. On main map: very hard to see where North is: could you make this clearer? I find it quite counterintuitive to not have the north points upwards, as is the case in other studies focusing on this glacier (e.g. see figs. 1b and 2 in Rabatel et al., 2018, Frontiers in Earth Science).

Similar to Figure 6, we have added a small inset showing the location of Glacier D'Argentiere within the European Alps. The N arrow has been changed in both maps to be more visible, and the full-glacier inset has been flipped such that N points upwards.

- l. 250: "Validating GIV...": you cannot really 'validate' your model by 'observing' dynamics of a glacier. I would rather refer to this as an evaluation, and would in fact suggest to simply remove this part to be consistent with other section heading: i.e. renaming this to '3.3 Vavilov ice dynamics'

We have changed the word 'validate' to 'evaluate' throughout, and agree about the limitations of this comparison as a 'validation'. We nevertheless believe the comparison is useful for readers (and that a full comparison between various feature tracking codes and ground based data should be the object of a future study). We have renamed the section according to you suggestion.

- l.252: You refer to 'Arctic land-ice' here: not clear if this includes the Greenland ice sheet (which is also Arctic land-ice...) or not, as seems to be the case when reading the next sentence (if so: be more specific and refer to the glaciers and ice caps explicitly)

We have used the terminology from the associated reference (Box et al., 2018) who include the Greenland Ice Sheet in this assessment. We have added a few words to the sentence to clarify this terminology.

- l. 255-266: many of the elements you mention here are really based on our knowledge of glacier surge: would be good to also refer to more general which this is explained, such as Sevestre and Benn (2015, JGlac; which also explains the phenomenon of glacier surging for glaciers in the Russian Arctic)

We have added a reference to Sevestre and Benn's paper in this section for readers searching for more background on surge type glaciers in general. Willis et al and Zheng et al's papers on Vavilov ice cap specifically also provide a nice summary of these processes.

- figure 8: same remark as for figure 6 and 7 for the location map. Why did you choose this color bar for the velocities? Very difficult to interpret and seems to be a color scheme that is meant to plot landscapes in fact.. Please change the color scheme to a more classic one that is meant for sequential data! In the caption of the figure, remove 'present', 'displays' and 'present'.

We have modified the color scheme to the same as Fig 7 (ColorBrewer YlGnBu, which should be both print and colorblind friendly). We have also added a small inlay of the broader region (October Revolution Island) and modified the caption as suggested.

- l. 283: '...if associated changed in...' should be '...if associated changes in...'

We have corrected the text.

- l. 284: ...'whether a similar peak occurs in 2020': not up to date anymore: incorporate 2020 in your explanation / figure above, or change the text here, possibly referring to 2021 instead of 2020.

We have updated the text to read 'in subsequent years'.

- l. 286: 'Method validation': comparing your results to those from another study is not a validation, but rather an evaluation. Moreover, it may be questioned whether you can evaluate your results by comparing them to another product which also has substantial uncertainties and potential large artefact (as you also mention, in e.g. l. 297-298). A real evaluation would for instance compare the GIV velocities to on-site high-precision velocity measurements based on GPS. Suggest removing the subsections 3.3.1 and 3.3.2 here and having the entire explanation under the section 3.3.

As mentioned in the comment above, we agree with the limitations of this comparison, and have reworded it as an 'evaluation'. We have also remove the subsections as recommended.

- l.300: 'Many tropical glaciers and ice caps have limited to no ice-flow data': well, they have a lot of data based on techniques such as the ones you present here. So be more specific here, e.g.: '...have limited to no ice-flow data from direct field measurements'

That is a good point, we have corrected the sentence to "Many tropical glaciers have limited to no ice-flow data from direct field measurements…"

- l.303: making the bridge from ice velocity to 'practical decision making' is quite a big step: how are these connected? Or maybe reword to: '...provide information on glacier state, which can contribute...'

We have adjusted the sentence as you recommend. This link was clearer in an older version of this manuscript, which included an inversion for ice thickness and volume.

- figure 9: remove the 'shows' and 'is' from the caption.

We have edited the caption to remove these words.

- l. 305: "...capped with an ice cap": maybe "...covered with an ice cap"

We have changed this sentence to "covered with 17 glaciers", as per the comment above about ice cap terminology.

- l. 317: you mention the runtime for this simulation, but you did not mention it for the other examples. Be consistent. Ideally, summarize all the runtime information in a single table, which could be added to the suppl. mat. Would also be useful, as directly after this (start of Discussion) you mention the computational aspect.

We have removed the mention of runtime. This value is computer-dependant, and is displayed by the toolbox itself at the start of a run (i.e. GIV will let users know how long a given run will take).

- l. 328-340: summary of other feature tracking algorithms: this information should appear earlier, in the introduction. In the discussion section, you should really focus on... the discussion (l.341-349), instead of giving a long summary of existing toolboxes. In fact, the real discussion is now very short (as is the conclusion): suggest merging the discussion and the conclusion in a single section.

We have moved a modified version of this discussion about other toolboxes to the introduction, and combined the discussion and conclusions section. Please see the tracked changes version of the manuscript for the full differences.

- l. 376: mention that this is a PhD thesis

We have updated this reference.

Thanks a lot for going through this list of comments and updating the manuscript accordingly. I look forward to receiving a new version of the manuscript, which could then be considered for final acceptance.

Once again, many thanks for all of the comments (and support for our choice to publish in The Cryosphere). We hope the substantial changes and supplements which we have added are also of use for other glaciologists and remote sensors working in this field.

Also, I just want to express my appreciation for the open review and pre-print publication approach taken by The Cryosphere (and similar journals). It has been very welcome to be able to share an earlier version of this paper with colleagues while it is in review, and the associated article metrics show that just over 1000 people have read the article online! This is always encouraging to see while working on revisions and edits, and I hope can widen the audience of this toolbox.

Many thanks,

M. Van Wyk de Vries and A. D. Wickert

Best regards,

Harry